# Environmental sensitivities of shallow-cumulus dilution. Part I: Selected thermodynamic conditions

Sonja Drueke[1], Daniel J. Kirshbaum[1], and Pavlos Kollias[2]

[1]Department of Atmospheric and Oceanic Sciences, McGill University, Montreal, QC, Canada
[2]School of Marine and Atmospheric Sciences, Stony Brook University, Stony Brook, NY, USA

**Correspondence:** Sonja Drueke (sonja.drueke@mail.mcgill.ca)

**Abstract.** Cumulus entrainment, and its consequent dilution of buoyant cloud cores, strongly regulates the life cycle of shallow cumuli yet remains poorly understood. Herein, new insights into this problem are obtained through large-eddy simulations that systematically investigate the sensitivity of shallow-cumulus dilution to cloud-layer relative humidity (RH), cloud- and subcloud-layer depths, and continentality (i.e., the land-ocean contrast). The simulated cloud-core dilution is found to be strongly sensitive to continentality, with fractional dilution rates twice as large over the ocean as over land. Using a similarity theory based on the turbulent-kinetic-energy (TKE) budget, the reduced cloud-core dilution over land is attributed to larger cloud-base mass flux ($m_b$), driven by stronger surface heating and subcloud turbulence. As $m_b$ increases, the fractional dilution rate must decrease to maintain energetic equilibrium. A positive sensitivity is also found to cloud-layer RH, with the core dilution increasing by 25-50 % for a 10 % enhancement in RH. This sensitivity is interpreted using the buoyancy-sorting hypothesis, in that mixtures of cloud and environmental air are more likely to become negatively buoyant and detrain (rather than diluting the cloud core) in drier cloud layers. By contrast, the sensitivities of (marine) shallow-cumulus dilution to cloud- and subcloud-layer depths are weak, with a 3 % decrease for a doubling for the former and a 4 % reduction in dilution for a 50 % deeper subcloud layer. These surprisingly weak sensitivities are readily explained by offsetting effects in the TKE similarity theory. Altogether, these experimental findings provide useful, though still incomplete, guidance for flow-dependent shallow-cumulus entrainment formulations in large-scale models.

## 1 Introduction

Shallow cumuli are ubiquitous over the subtropical oceans (with a frequency of occurrence of 10-30 %; e.g., Norris, 1998) and during the warm season over land. Despite their small sizes, short lifetimes, and small cloud fractions, they play an essential role in weather and climate by regulating the thermodynamic and kinematic structure of the lower-to-middle troposphere. An important process affecting the development of such clouds is entrainment, by which environmental air is ingested into the clouds. Among other things, entrainment causes the evaporation of cloud droplets, which reduces the liquid water content (LWC), buoyancy, and vigor of convection (e.g., Derbyshire et al., 2004; Gerber et al., 2008; Krueger, 2008; Del Genio, 2012; Lu et al., 2013).

Cumulus entrainment is caused by both microscale turbulent mixing along the cloud perimeter ("turbulent entrainment") and cloud-scale circulations drawing organized inflow ("dynamic entrainment") (e.g., Houghton and Cramer, 1951; de Rooy et al., 2013). Both lead to cloud dilution, or the change of internal cloud properties due to cloud-environmental mixing. Entrainment and dilution would be equal if the entrained air was drawn directly from the environment and retained within the cloud. However, cumuli are not surrounded by pure environmental air but rather by subsiding "shells" containing mixtures of cloud and environmental air (e.g., Heus and Junker, 2008). Entrainment of recycled cloudy air from the cloud shell tends to limit the degree of dilution for a given amount of entrainment (e.g., Zhang et al., 2016). Also, cloud-environmental mixtures may become negatively buoyant and detrain rather than diluting the cloud core. The process of rejecting negatively buoyant air leads to a "concentration" of buoyancy within the cloud core.

Two fundamentally different methods have been used to diagnose entrainment: "bulk" methods infer entrainment based on conditionally averaged profiles of conserved variable(s) (e.g., Betts, 1975; de Rooy et al., 2013) and "direct" methods calculate the flux of air across the cloud surface (Romps, 2010; Dawe and Austin, 2011). While the former can be evaluated readily from both observations and numerical simulations, the latter can only be determined from intensive calculations on large-eddy simulations (LES). Thus, the bulk calculation is more flexible and efficient. However, the term "bulk entrainment" is misleading because, based on the above definitions, this method actually quantifies cloud *dilution*, and will thus be termed "bulk dilution" herein. In LES experiments, Hannah (2017) found only a weak statistical correlation existed between direct entrainment and bulk dilution, reflecting their very different physical meanings.

The evolution of cumuli in general, and entrainment and dilution in particular, are sensitive to environmental conditions such as cloud-layer humidity (e.g., Wang and McFarquhar, 2008; Derbyshire et al., 2011; Lu et al., 2018; Bera and Prabha, 2019), static stability (e.g., Wang and McFarquhar, 2008; Tian and Kuang, 2016), aerosol loading (e.g., Jiang et al., 2006; Wang and McFarquhar, 2008; Small et al., 2009; Seigel, 2014), surface heterogeneities (e.g., Rieck et al., 2014), and vertical wind shear (e.g., Brown, 1999; Lin, 1999). Among these diverse factors, we focus herein on the sensitivities of shallow-cumulus dilution to thermodynamic conditions, with the sensitivity to the vertical wind profile deferred to a companion paper.

Numerous studies have investigated the impact of cloud-layer relative humidity on cumulus entrainment/dilution. When entrained into cloud cores, drier air causes more evaporation and buoyancy loss, which tends to suppress convection (e.g., Stommel, 1947; Brown and Zhang, 1997; Sherwood, 1999; Holloway and Neelin, 2009). Using a cloud-resolving model (CRM), Derbyshire et al. (2004) investigated the sensitivity of deep convection to cloud-layer humidity, finding reduced cloud-updraft mass fluxes in drier environments. Bulk-entrainment profiles from these same simulations in de Rooy et al. (2013) revealed a systematic increase in cloud dilution within drier environments.

In contrast, LES of shallow cumuli over the Indian Ocean by Wang and McFarquhar (2008) indicated a positive sensitivity of dilution to relative humidity (RH). Similar results were obtained in CRM simulations of continental cumuli (Stirling and Stratton, 2012) and LES of monsoon cumuli (Bera and Prabha, 2019) as well as by aircraft observations from the Rain in Cumulus over the Ocean (RICO) and Routine AAF CLOWD Optical Radiative Observations (RACORO) field campaigns (Lu et al., 2018). The latter study suggested that higher RH leads to reduced cloud-core buoyancy and lower updraft speeds, and hence increased dilution. On the other hand, Tian and Kuang (2016) found that neither cloud-layer RH nor environmental

stratification strongly impacted simulated cumulus dilution. Altogether, no consensus has been reached on the sensitivity of cloud dilution to cloud-layer RH.

A wide range of cumulus dilution rates have been reported over different regions. In observations of trade-wind cumuli over the Pacific and Atlantic oceans, cloud dilution rates of about 1.3 km$^{-1}$ were reported (Raga et al., 1990; Gerber et al., 2008). Similar dilution rates were found in LES of maritime trade-wind cumuli (Siebesma, 1998; Stevens et al., 2001; vanZanten et al., 2011). Over the Indian Ocean, however, Wang and McFarquhar (2008) found smaller dilution rates of about 0.9 km$^{-1}$, which are similar to the ones obtained by Brown et al. (2002) for simulated continental cumuli over the US Department of Energy's Atmospheric Radiation Measurement (ARM) Southern Great Plains (SGP) observatory in Oklahoma. Over the Florida peninsula, Neggers et al. (2003a) found bulk-dilution rates of 1.5-3.0 km$^{-1}$ both in observations and LES. Although trends cannot be easily inferred from such a small observational sampling, these findings suggest geographic differences in cloud dilution, possibly related to whether the clouds form over land or the ocean. Although a potential sensitivity to continentality was recognized by Brown et al. (2002), it has not yet been explored in detail or physically interpreted.

One of the factors often invoked to explain cumulus dilution rate is horizontal cloud size. Assuming a cylindrical cloud with homogeneous entrainment, the rate of cloud dilution scales inversely with cloud radius ($R$; e.g., de Rooy et al., 2013). This sensitivity arises because, at a given level, the flux of entrained air scales with the cloud perimeter ($\sim R$) but the entrained air mixes over the cloud area ($\sim R^2$), implying less dilution at larger $R$. Although entrainment is far from homogeneous in real clouds, both observational and LES studies indicate weaker dilution for larger cloud horizontal areas (e.g., Squires and Turner, 1962; Dawe and Austin, 2012; Khairoutdinov and Randall, 2006; Kirshbaum and Grant, 2012; Rousseau-Rizzi et al., 2017).

Cloud size is determined by numerous factors, one being the cloud-layer depth. In 3D isotropic turbulence, the horizontal size of turbulent eddies broadly scales with the layer depth. Thus, deeper cloud layers may support wider clouds and consequently smaller dilution rates. Using CRM simulations of the diurnal cycle of convection over land, Del Genio and Wu (2010) showed that the dilution rates of shallow cumuli exceeded those for cumulonimbi. While subsequent studies have reached similar conclusions (e.g., Derbyshire et al., 2011; Stirling and Stratton, 2012), little attention has been paid to the sensitivity of cloud size and dilution to cloud-layer depth within the shallow-cumulus regime alone.

The subcloud layer may also influence cloud size, mainly by regulating the size of turbulent eddies that breach the lifting condensation level to form clouds. Cloud-resolving numerical simulations have shown that larger incipient thermals at the level of free convection (LFC) evolve into less dilute and deeper clouds (e.g., Rousseau-Rizzi et al., 2017). One factor regulating initial cloud size is the subcloud layer depth; deeper subcloud layers give rise to wider thermals that tend to initiate wider clouds. To date, however, no study has systematically examined the sensitivity of shallow-cumulus dilution to subcloud-layer depth. Another factor regulating initial cloud size is the characteristic scale of surface heterogeneities. The idealized LES of Rieck et al. (2014) showed that, through their control over cloud size, such heterogeneities also regulated cumulus dilution within the cloud layer.

The above discussion indicates that, while the controls on shallow-cumulus entrainment and dilution are an active area of research, at least some of the environmental sensitivities and corresponding physical explanations of this process are either poorly understood or differ between different studies. Improving this understanding is essential given that the representation of

cumulus convection, and particularly the entrainment process, is a major source of global climate sensitivity in those models
(e.g., Murphy et al., 2004; Knight et al., 2007; Sanderson et al., 2008; Rougier et al., 2009; Klocke et al., 2011). In both global climate and many weather forecast models, shallow cumuli are parameterized due to insufficient grid resolution, and will continue to be parameterized for some time.

The simplest approach to parameterizing entrainment is to prescribe a fixed entrainment profile within the cloud layer. However, this method neglects the wide range of entrainment/dilution rates found in reality. Raymond and Blyth (1986) and Kain
and Fritsch (1990) addressed this limitation by invoking the "buoyancy-sorting" concept, where entrainment and detrainment are determined by the buoyancy of cloud-environmental mixtures. Positively buoyant mixtures are retained within the core and added to the entrainment rate, while negatively buoyant mixtures are expelled from the cloud and contribute to detrainment. This method links dilution to convective available potential energy (CAPE) and cloud-layer RH but ignores other potentially important parameters like continentality and vertical wind shear. Alternatively, some parameterizations link entrainment to pa-
rameterized updraft speed (e.g., Neggers et al., 2002), buoyancy (e.g., von Salzen and McFarlane, 2002), or both (e.g., Gregory, 2001). Furthermore, while recent entrainment formulations have directly accounted for the cloud-layer RH sensitivity, the sign of this sensitivity differs in different schemes (e.g., Bechtold et al., 2008; Stirling and Stratton, 2012), reflecting the incomplete understanding of this sensitivity.

To improve the representation of convection in global models, the understanding of shallow-cumulus dilution must improve.
The present study focuses on the sensitivities of shallow-cumulus dilution to certain environmental parameters that, as indicated in the above discussion, either remain unclear or have not received adequate attention. These include relative humidity, continentality, subcloud-layer depth, and cloud-layer depth. To help resolve contradictory findings from past studies, we use a consistent LES framework, applied to a broad range of continental and maritime shallow-cumulus cloud fields, to systematically examine and physically interpret each sensitivity. The numerical setup and the different test cases are detailed in Sect. 2,
along with a description of the analysis methods. In Sect. 3, the sensitivities of the fractional dilution rate to thermodynamic profiles and the continentality are presented. A discussion of these results is presented in Sect. 4. Section 5 concludes this study.

## 2 Methodology

### 2.1 LES of shallow cumulus cloud fields

For the numerical experiments, we use the Bryan Cloud Model version 17 (CM1; Bryan and Fritsch, 2002), a fully nonlinear, compressible, and nonhydrostatic atmospheric model. A monotonic 5th-order Weighted Essentially Non-Oscillatory (WENO) advection scheme is used for both scalars and velocity vectors. For time integration, a third-order Runge-Kutta scheme with a split time step to account for acoustic waves is employed. All simulations use a $f$-plane approximation and have periodic horizontal, semi-slip lower, and free-slip upper boundary conditions. A Rayleigh damping zone of 500-m depth immediately
below the model top is used to limit the spurious reflection of internal-gravity waves at the upper boundary. Further numerical settings depend on the individual cases and are listed in Table 1.

Four different shallow-cumulus ensembles form the basis of this study, covering different environmental conditions in which shallow cumuli develop in nature. Of these cases, two are based on field campaigns that were conducted in maritime environments in the eastern Caribbean Sea, whereas the other two field experiments were conducted in the continental environment at the ARM-SGP central facility in northern Oklahoma. These field campaigns are the Barbados Oceanographic and Meteorological Experiment (BOMEX; Holland and Rasmusson, 1973), the Rain in (Shallow) Cumulus over the Ocean (RICO; Rauber et al., 2007) study, a LES comparison study of the diurnal cycle of shallow cumulus convection at the ARM-SGP observatory (ARM-SGP; Brown et al., 2002) and the RACORO campaign (also at the SGP observatory; Vogelmann et al., 2012).

Based on the field observations, LES intercomparison studies described in Siebesma et al. (2003) for BOMEX, vanZanten et al. (2011) for RICO, and Brown et al. (2002) for ARM-SGP and an observation-LES study by Endo et al. (2015) for RACORO were undertaken and provide the configurations for our LES experiments (Table 1). All simulations are initialized with vertical profiles of potential temperature ($\theta$), water-vapor mixing ratio ($q_\mathrm{v}$), and horizontal winds ($u$ and $v$) as prescribed in the above-mentioned studies, along with small-amplitude random perturbations in $\theta$ and $q_v$ to seed convective motions. Most simulations are conducted with the grid spacings mentioned in the reference literature (ranging from 64 m to 150 m). Additional high-resolution simulations with double the resolution have been conducted (indicated by the parenthetic grid spacings in Table 1).

Large-scale subsidence, horizontal advection of heat and moisture, longwave cooling, and surface fluxes are adopted directly from Siebesma et al. (2003) (BOMEX), vanZanten et al. (2011) (RICO), Brown et al. (2002) (ARM-SGP), and Endo (personal communication; RACORO). The surface heat fluxes for the two maritime cases are prescribed (BOMEX) or calculated interactively (RICO) and vary only slightly in time in the latter. The surface fluxes are prescribed for both continental cases and exhibit a strong diurnal cycle. The cloud droplet number concentrations, which are fixed within each simulation, are specified with smaller values for the maritime cases (100 cm$^{-3}$ for BOMEX and 70 cm$^{-3}$ for RICO) than for the continental cases ARM-SGP (250 cm$^{-3}$) and RACORO (500 cm$^{-3}$). For further details on the configurations of these cases, we refer the reader to the aforementioned studies. These simulations reproduce the characteristics of the shallow-cumulus cloud fields presented in the corresponding LES intercomparison studies (see Fig. 1 in Drueke et al., 2019).

## 2.2 Analysis methods

### 2.2.1 Bulk-dilution rate

The simulated fractional dilution rate is calculated following Siebesma and Cuijpers (1995, hereafter SC95), where a formulation of the prognostic equation of conserved variables such as total water specific humidity ($s_\mathrm{t}$) is utilized. By decomposing the equation for $s_\mathrm{t}$ into a cloud "core" and an environmental part, SC95 obtained

$$E\left(s_{\mathrm{t_{env}}} - s_{\mathrm{t_{co}}}\right) = M_{\mathrm{co}}\frac{\partial s_{\mathrm{t_{co}}}}{\partial z} + \frac{\partial a_{\mathrm{co}}\overline{\rho w' s_\mathrm{t}'}^{\mathrm{co}}}{\partial z} + a_{\mathrm{co}}\rho\frac{\partial s_{\mathrm{t_{co}}}}{\partial t} - a_{\mathrm{co}}\rho\left(\frac{\partial \overline{s_\mathrm{t}}}{\partial t}\right)_{\mathrm{forcing}}, \qquad (1)$$

where $E$ is the dilution rate and 'co' denotes the cloud core. The plain overbar is a horizontal domain average and the overbar indexed 'co' is a conditional average over the cloud cores (with respect to the cloud-core average). At a given vertical level,

the cloud-core properties are calculated as the conditional average over all horizontal grid points that contain significant liquid water ($q_c > 0.01$ g kg$^{-1}$), are positively buoyant with respect to the horizontal domain average and are ascending, similar to Siebesma et al. (2003). We define the area-averaged cloud-core mass flux as $M_{co} \equiv \overline{\rho} a_{co} w_{co}$, where $\rho$ is the air density, $a_{co}$ is the cloud-core fraction, and $w_{co}$ is the conditionally averaged cloud-core vertical velocity. The fractional dilution rate is obtained by dividing Eq. (1) by $M_{co}$ ($\varepsilon_{SC95} = E/M_{co}$). We calculate $\varepsilon_{SC95}$ at each model output time, over all vertical levels where cloud cores are simulated, to obtain instantaneous vertical $\varepsilon_{SC95}$ profiles.

While entrainment describes the ingestion of environmental air into the cloud, detrainment refers to the expulsion of cloudy air into the environment. The detrainment rate $D$ can be determined by using the obtained dilution rate and the continuity equation (SC95) to give

$$\rho \frac{\partial a_{co}}{\partial t} = -\frac{\partial M_{co}}{\partial z} + E - D . \tag{2}$$

The fractional detrainment rate is obtained by dividing Eq. (2) by $M_{co}$ ($\delta_{SC95} = D/M_{co}$).

### 2.2.2   TKE similarity theory

To gain physical insight into the sensitivities of the diagnosed $\varepsilon_{SC95}$ to environmental conditions, we use the shallow-cumulus similarity theory based on the turbulent-kinetic-energy (TKE) budget of Grant and Brown (1999) and Grant and Lock (2004). This theory allows for the estimation of $\varepsilon$ and other bulk properties based on limited information about the larger-scale environment and the subcloud layer. To the extent that it captures the simulated sensitivities of cloud dilution environmental conditions, it can help to determine the underlying physical causes.

For non-precipitating, weakly sheared shallow-cumulus-cloud fields in statistical equilibrium, the ensemble-averaged rate of buoyancy production of TKE ($\overline{w'b'}$) and the turbulent dissipation ($d$) dominate the TKE budget

$$\overline{w'b'} \approx d , \tag{3}$$

where $w$ is vertical velocity and $b$ is buoyancy. Perturbations from the horizontally averaged mean are indicated by primes. Through a scale analysis, Grant and Lock (2004) inferred that

$$\overline{w'b'} \approx \left( \frac{m_b}{w^*} \right)^{1/2} m_b \frac{CAPE}{z_{cld}} , \tag{4}$$

where $m_b$ is the density-normalized cloud-core-base mass flux, $w^*$ is the turbulent vertical-velocity scale, CAPE is the convective available potential energy of the cloud-core layer, and $z_{cld}$ is the cloud-core-layer depth. The nondimensional factor $(m_b/w^*)^{1/2}$ can be interpreted as a cloud fractional area (Grant and Lock, 2004). The dissipation is scaled as

$$d \approx \left( \frac{m_b}{w^*} \right)^{1/2} \frac{w^{*3}}{z_{cld}} , \tag{5}$$

and, using Eqs. (3)-(5),

$$w^* \approx (m_b CAPE)^{1/3} . \tag{6}$$

To scale the cloud dilution, two assumptions are made: (i) entrainment is an intermediate process between buoyancy production and turbulent dissipation, and (ii) a fixed fraction ($A_\varepsilon$) of updraft kinetic energy is imparted to the portion of entrained air that reaches and dilutes the cloud core. The kinetic energy transferred to the core-entrained air, or $\varepsilon m_b w^{*2}$, is then multiplied by the scaling parameter $(m_b/w^*)^{1/2}$ (like the other TKE source terms above) and equated to $A_\varepsilon d$:

$$\left(\frac{m_b}{w^*}\right)^{1/2} \varepsilon m_b w^{*2} \approx A_\varepsilon \left(\frac{m_b}{w^*}\right)^{1/2} \frac{w^{*3}}{z_{cld}} \; . \tag{7}$$

Substituting Eq. (6) into Eq. (7), we obtain

$$\varepsilon_{TKE} = A_\varepsilon \frac{CAPE^{1/3}}{m_b^{2/3}} \frac{1}{z_{cld}} \; . \tag{8}$$

Thus, based on this theory, the fractional dilution rate ($\varepsilon$) increases weakly with CAPE and more strongly decreases with increasing cloud depth and cloud-base mass flux. Estimations of $A_\varepsilon$ in LES of maritime shallow cumuli have ranged from 0.03-0.06 (Grant and Lock, 2004; Kirshbaum and Grant, 2012). Over a larger sampling of both maritime and continental cases, Drueke et al. (2019) estimated $A_\varepsilon = 0.035$.

Although the TKE similarity theory has impressively captured sensitivities of cloud dilution in different studies (Kirshbaum and Grant, 2012; Drueke et al., 2019), it has several limitations of note, one being the assumption of statistical equilibrium. While few real-world cloud fields strictly meet this condition, the associated errors are small even in rapidly evolving cloud fields (Kirshbaum and Grant, 2012). Furthermore, Drueke et al. (2019) showed that the theory, when used as a simulated cloud-dilution retrieval, reasonably estimated dilution rates for both maritime and diurnally forced continental cloud ensembles, the latter of which are inherently transient. In addition, this theory does not account for the impacts on environmental RH on cloud dilution, and thus cannot be expected to explain this sensitivity. The assumption of a negligible shear production/loss term also restricts the applicability of this theory to weakly sheared cloud fields. Finally, the two above-mentioned assumptions (that dilution scales with other TKE source terms and $A_\varepsilon$ is fixed) may limit the accuracy of the resulting dilution estimates.

## 3   Results

To begin, we examine the evolution of seven quantities—sensible and latent heat fluxes ($H$ and $LE$, respectively), total cloud cover ($cc$), cloud-base and cloud-top heights ($z_b$ and $z_t$, respectively), cloud-base mass flux ($m_b$), and vertically integrated and horizontally averaged TKE—for one maritime (BOMEX) and one continental (ARM-SGP) case in Fig. 1. For BOMEX, the sensible and latent heat fluxes are constant in time (Fig. 1a) and, after an initial spin-up of about two hours, the cloud ensemble reaches a statistical quasi-steady state. The cloud cover, defined as the fraction of vertical columns containing at least one grid point with $q_c > 0.01$ g kg$^{-1}$, maintains a low value of around 10-12 %, and the cloud-base and cloud-top height are roughly constant at ~0.5 km and ~2.0 km, respectively. The $m_b$ fluctuates around 0.04 m s$^{-1}$ throughout the simulation, and, after some initial transience, the TKE remains roughly constant at around 350 m$^3$ s$^{-2}$ over the last 4 h (left column in Fig. 1). The second maritime case (RICO; not shown) is also characterized by a cloud-base height of 0.5-0.6 km, but the cloud layer is

deeper and capped by an inversion based at around 2.0 km, which fosters sufficiently deep convection to produce precipitation with a domain-averaged rain rate of 0.18 mm h$^{-1}$ over the final 8 h. Little to no precipitation forms in the other cases.

The continental ARM-SGP case undergoes a pronounced diurnal cycle (right column in Fig. 1 for ARM-SGP; RACORO is not shown). Clouds initiate in the mid-morning with an increasing total cloud cover that reaches a maximum of about 25 % in the early afternoon and decreases toward the evening (Fig. 1d). Due to the warming, drying, and vertical growth of the subcloud layer, the cloud base rises from initially ∼0.7 km in the mid-morning to ∼1.3 km in the early evening. The cloud-top height reaches a maximum of ∼2.9 km in the late afternoon (Fig. 1f). Unlike the roughly constant $m_{\mathrm{b}}$ in BOMEX and RICO, $m_{\mathrm{b}}$ is zero for several hours before rising rapidly to a maximum at around 4 h (greatly exceeding the maximum in BOMEX) and then decreasing. The TKE increases more gradually to a maximum about six times that of BOMEX at around 4 h, then decreases. Each afternoon of the RACORO three-day simulation evolves similarly to that of ARM-SGP, except that different days exhibit different levels of cloudiness due to variations in large-scale forcing (not shown).

The above four cases capture a range of environmental conditions in which shallow cumuli develop and form the basis of our analysis. However, since several environmental aspects vary between these cases, the attribution of dilution sensitivity to any specific parameter would be difficult with these experiments alone. Therefore, to isolate environmental parameters of interest, we perform additional experiments with initial conditions systematically varied in one or more cases. A summary of the full set of numerical experiments is given in Table 2.

The experiments are grouped according to the environmental parameters under consideration and are explained in further detail in the subsections below. For all cases, "control" (CTRL) simulations with initial conditions and numerical configurations drawn from the referenced literature have been conducted (Tables 1 and 2). We also rerun each CTRL case at twice the horizontal grid resolution (CTRL-HR). Six repetitions of each experiment with different fields of random initial perturbations are used to roughly estimate the expected value and uncertainty of the results.

To determine the cloud dilution for each experiment, we calculate 15-min running averages of $\varepsilon_{\mathrm{SC95}}$ profiles, from which bulk $\varepsilon_{\mathrm{SC95}}$ averages over the central 50 % of the cloud layer are obtained. This central section of the cloud layer is chosen to exclude the effects of organized inflow and outflow through the cloud base and top, respectively. While the choice of the averaging depth is somewhat arbitrary, tests with averages over 30 % and 70 % of the central cloud layer only modestly changed the results (Drueke et al., 2019). The bulk $\varepsilon_{\mathrm{SC95}}$ values as well as the vertical $\varepsilon_{\mathrm{SC95}}$ profiles themselves are further averaged over a selected 3-hr period for all cases and environmental conditions. For the maritime cases with nearly statistically steady cloud ensembles, the averages are calculated over the last 3 h of the simulations. Because the clouds in the continental cases are most numerous in the afternoon, the averaging window runs from 13 to 16 local solar time (LST), as indicated by the gray shading in Fig. 1.

## 3.1 Cloud-layer RH

To examine the sensitivity of cloud dilution to cloud-layer RH, the initial profiles of cloud-layer RH for the BOMEX maritime case (Fig. 2a) and the ARM-SGP continental case (Fig. 2b) are systematically modified. The modifications are applied to the cloud-bearing layer just above the subcloud layer, which extends from 0.5-1.5 km in BOMEX and from 0.7-1.3 km in ARM-

SGP. The saturation RH deficit at the top of the cloud layer is halved for the more humid case (MOIST1), increased by the same amount for the drier case (DRY1), and increased by twice that amount for the driest case (DRY2). Within this layer, $q_v$ varies linearly between between its cloud-base value and its modified value at the layer top. The increased $q_v$ at the cloud-layer top is maintained up to the top of the sounding.

The initial variation of RH in the prescribed initial state is not fully maintained over the course of the simulations. In BOMEX, where only the CTRL simulations are initialized in quasi-equilibrium, the horizontally averaged RH profiles in the drier and moister cases both drift slowly toward the CTRL profile over time (not shown). As a result, the range of mean cloud-layer RH (measured over the central 50 % of the layer) evolves from 76-95 % (from DRY2 to MOIST1) at the initial time to 77-90 % over the 3-h analysis window. In ARM-SGP, the range of RH also decreases modestly from 74-87 % initially to 77-87 % over the 3-h analysis window. Thus, although the RH gap between the DRY2 and MOIST1 cases narrows over time, the majority of the initial range is maintained through the analysis period. To account for this slight model drift, the mean cloud-layer RH values reported below correspond to the 3-h analysis period, not the initial state.

In both BOMEX and ARM-SGP, the dilution rate ($\varepsilon_{\mathrm{SC95}}$) is positively correlated with cloud-layer RH (Figs. 3a and b). The range of $\varepsilon_{\mathrm{SC95}}$ for the six different realizations of each experiment, as indicated by the error bars in Fig. 3, is smaller than the trend induced by cloud-layer RH variations. A 10 %-increase in RH leads to large dilution increases of about 25 % (BOMEX) and 47 % (ARM-SGP). This finding is consistent with Lu et al. (2018), who interpreted the positive correlation between $\varepsilon$ and RH using the buoyancy-sorting concept. Mixtures of cloud and environmental air with lower RH are more likely to become negatively buoyant and detrain without diluting the cloud core. The correspondingly larger detrainment rates ($\delta_{\mathrm{SC95}}$) found in drier environments support this explanation (Figs. 3c and d). In environments with higher RH, fewer entrained parcels become negatively buoyant, and thus a higher fraction of entrained air remains within the cloud core, leading to increased dilution.

The difference between the fractional dilution and detrainment rates ($\varepsilon - \delta$) regulates the normalized vertical gradient of the convective mass flux (e.g., Betts, 1975). Since $\delta$ generally exceeds $\varepsilon$ for all experiments (Figs. 3e and f), this difference is negative and implies a diminishing cloud mass flux with height (Fig. 4a and c). This difference is maximized in drier environments (consistent with Wang and McFarquhar, 2008), and hence the cumuli remain smaller and narrow more rapidly with height in those cases. This narrowing is demonstrated by profiles of effective cloud radius ($R_{\mathrm{eff}}$), defined at a given level as

$$R_{\mathrm{eff}} = \frac{1}{N_{\mathrm{c}}} \sum_{i=1}^{N_{\mathrm{c}}} \left( \frac{N_{\mathrm{g}_i} \Delta_{\mathrm{x}} \Delta_{\mathrm{y}}}{\pi} \right)^{1/2} . \tag{9}$$

where $N_{\mathrm{c}}$ is the number of clouds at that height and $N_{\mathrm{g}}$ the number of horizontal grid points in each cloud (Fig. 4b and d). The effective radius is the smallest in the driest versions of both the BOMEX and ARM-SGP cases. The slight increase in cloud-base height in drier environments stems from the mixing of drier cloud-layer air into the subcloud layer. A schematic of the mechanisms underlying the cloud-layer RH sensitivity is given in Fig. 5.

## 3.2 Layer depth

To investigate the dependence of $\varepsilon$ on the cloud-layer and subcloud-layer depth, we focus exclusively on the nearly statistically
steady BOMEX case. The strong diurnal forcing in ARM-SGP renders the layer depths transient and difficult to control.

### 3.2.1 Cloud-layer depth

We vary the initial cloud-layer depth, $(z_{\text{cld}})_0$, by raising or lowering the base of the trade-wind inversion from the CTRL case
with the $\theta$ lapse rate held fixed in each layer. Three experiments are conducted, one with a 250-m shallower cloud layer than
CTRL (CL-SHAL1), one with a 250-m deeper cloud layer than CTRL (CL-DEEP1), and one with a 500-m deeper cloud layer
than CTRL (CL-DEEP2) (Fig. 6a). To maintain quasi-equilibrium, the profiles of the large-scale forcings within the cloud layer
(subsidence and longwave radiative cooling) are shortened or lengthened accordingly.

In any shallow cloud ensemble, the clouds may exhibit a wide range of depths at any instant, with most cloud tops falling well
below the cloud-layer top. The distribution of individual cloud depths, normalized by the cloud-layer depth, is a morphological
property that can be compared between different cases to assess their level of dynamic similarity. Systematic differences in
these distributions would indicate that individual cloud depths do not simply scale with the cloud-layer depth (in a statistical
sense), which could complicate a direct comparison of cloud dilution between them. To evaluate the similarity of the cloud
ensembles for the current sensitivity tests, we compare their normalized-cloud-depth probability density functions (PDF) in
Fig. 6b. At any given time, the vast majority of clouds have depths ($z_{\text{d}}$) of less than half the cloud-layer depth ($\overline{z}_{\text{d}}$), and very
few reach the cloud-layer top. The distribution of normalized cloud depths is similar in all four cases, suggesting that these
cloud ensembles can be directly compared on equal footing.

For each experiment, $\varepsilon_{\text{SC95}}$ profiles are diagnosed for each ensemble member using Eq. (1) and then averaged over all six
ensemble members, with the ensemble standard deviation shown by yellow shading (Fig. 6c). These profiles are similar over
the lowermost 500 m of the cloud layer and only diverge above $\sim$1.2 km, with more rapid vertical decay for the shallower
cloud layers. This apparently weak sensitivity is reflected by the layer-averaged $\varepsilon_{\text{SC95}}$ that decreases by 3 % as $(z_{\text{cld}})_0$ doubles
from 0.75 m (CL-SHAL1) to 1.5 m in (CL-DEEP2) (Fig. 7a and c).

To physically interpret the above dilution sensitivity (or lack thereof), we use the TKE similarity theory of Sect. 2.2.2 with
constant $A_\varepsilon = 0.035$ (as in Drueke et al., 2019). The value of $\varepsilon_{\text{TKE}}$ depends on three larger-scale parameters: $m_{\text{b}}$, $z_{\text{cld}}$, and
CAPE (Eq. (8)). The $m_{\text{b}}$ is evaluated at the lowest height at which $\overline{q_{\text{c}}}^{\text{co}} > 0.01$ g kg$^{-1}$, $z_{\text{cld}}$ is the depth of the cloud-core layer
where $\overline{q_{\text{c}}}^{\text{co}} > 0.01$ g kg$^{-1}$, and CAPE is integrated over this same layer. Evaluating Eq. (8) using these parameters reveals a
good match to the $\varepsilon_{\text{SC95}}$ diagnoses (Fig. 7a). While $\varepsilon_{\text{TKE}}$ agrees very well with $\varepsilon_{\text{SC95}}$ for the CTRL, CL-DEEP1, and CL-
DEEP2 experiments, it overestimates the dilution rate for the CL-SHAL1 by $\sim$0.1 km$^{-1}$. While this estimate lies outside the
one-standard-deviation range of $\varepsilon_{\text{SC95}}$, the relative error is still less than 10 %.

As shown in Fig. 7b, $m_{\text{b}}$ is similar for the four cases, which is expected because the initial subcloud conditions are held
fixed. Similar to the initial range of $(z_{\text{cld}})_0$ specified above (0.75-1.5 km), $z_{\text{cld}}$ during the 3-h analysis period ranges from
$\sim$0.95 m in CL-SHAL1 to $\sim$1.65 m in CL-DEEP2 (Fig. 7c). These increases in $z_{\text{cld}}$, which tend to decrease $\varepsilon_{\text{TKE}}$ in Eq. (8),

are approximately offset by increases in CAPE, the cube root of which increases similarly to $z_{\text{cld}}$ (Fig. 7d and e). As a result, $\varepsilon$ remains relatively constant as $(z_{\text{cld}})_0$ is varied.

### 3.2.2 Subcloud-layer depth

Analogous to the above experiments, we vary the initial subcloud-layer depths by raising (SCL-DEEP1) or lowering (SCL-SHAL1) the cloud base by 250 m from its value in CTRL, with $\theta$ and $q_v$ lapse rates again held fixed in each layer (Fig. 8a). To maintain a constant cloud-base RH for all experiments, we increase (SCL-SHAL1) or decrease (SCL-DEEP1) the subcloud $q_v$ accordingly. Also, we keep the cloud-layer vertical wind shear fixed at the CTRL value of 1.8 m s$^{-1}$ km$^{-1}$, with shear bases located 200 m above the cloud-layer bases in all cases. Finally, the prescribed profiles of large-scale subsidence, advection, and longwave radiative cooling are modified so that the forcings at the surface, cloud base, and cloud top are identical across the experiments.

Despite the adjustments to the large-scale forcing profiles to minimize the degree of disequilibrium in these cases, one of the two sensitivity tests (SCL-SHAL1) exhibits noticeable transience during the analysis period. Its cloud-base height increases from its initial, prescribed value of 250 m to an average of 350 m over the analysis period (not shown). By contrast, the cloud-base heights for CTRL and SCL-DEEP1 remain nearly fixed at their respective initial values of 500 m and 750 m. Although, for the sake of completeness, we show the results of SCL-SHAL1 in our subsequent analysis, its more transient nature may lead to a lack of robustness.

As before for the cloud-layer-depth experiments, we compare PDFs of normalized cloud depth for these three cases (Fig. 8b). The similar distributions thus produced suggests that the cloud ensembles are dynamically similar and can be straightforwardly compared. Near cloud base, the diagnosed $\varepsilon_{\text{SC95}}$ modestly but systematically decreases as the subcloud-layer depth is increased, while the value near cloud top remains similar (Fig. 8c). The layer-averaged $\varepsilon_{\text{SC95}}$ decreases by a total of about 15% for the near-tripling of the subcloud-layer depth between SCL-SHAL1 and SCL-DEEP1 (Fig. 9a). Although the transient SCL-SHAL1 case must be interpreted with caution, comparison between it and the CTRL case produces a similar trend as that found between CTRL and SCL-DEEP1.

As before, we use the TKE theory, as embodied in Eq. (8), to physically interpret the results. This theory reasonably captures the modest sensitivity of $\varepsilon_{\text{SC95}}$ to subcloud-layer depth (Fig.9a), even for the transient SCL-SHAL1 case. Similar to the offsetting tendencies in Sect. 3.2.1, a ~5% increase of $z_{\text{cld}}$ is compensated by a 5% increase of CAPE$^{1/3}$ for the CTRL and SCL-DEEP1 experiments (Figs.9c, d and e). For its part, $m_{\text{b}}$ tends to increase with subcloud-layer depth (Fig.9b), possibly owing to stronger, less hydrostatic turbulence in deeper subcloud layers (e.g., Tang and Kirshbaum, 2020). With offsetting effects on $z_{\text{cld}}$ and CAPE, the modest increase of $m_{\text{b}}$ in deeper subcloud layers explains a modest reduction in $\varepsilon_{\text{TKE}}$. An elaboration on the physical link between $\varepsilon_{\text{TKE}}$ and $m_{\text{b}}$ is provided below.

### 3.3 Continentality

Along with the aforementioned sensitivity of $\varepsilon$ to RH, a second sensitivity is apparent in Fig. 3: the dilution rates in the maritime BOMEX experiments are consistently larger than those in the continental ARM-SGP experiments. To more directly investigate

the impact of continentality on the dilution rate, we compare all four test cases, two of which are maritime (BOMEX and RICO) and two of which are continental (ARM-SGP and RACORO). For BOMEX, RICO, and ARM-SGP, we also consider variations (DRY1 and MOIST1 as well as CTRL-HR) to increase the sample size. Although cloud-layer RH variation is not considered in the RACORO case, a CTRL-HR simulation is performed and each of the three days of RACORO is counted as a separate case.

Based on these 18 experiments, a robust sensitivity of $\varepsilon_{SC95}$ to continentality is found, in that the cloud dilution is consistently larger for the maritime cloud fields than for the continental ones (Fig. 10a). Whereas $\varepsilon_{SC95}$ averages to 0.57 km$^{-1}$ over the continental cases, it averages to 1.24 km$^{-1}$ for the maritime cases, and hence $\varepsilon_{ocean} \approx 2.2 \times \varepsilon_{land}$. The TKE similarity theory gives similar values, with a mean $\varepsilon_{TKE}$ of 1.18 km$^{-1}$ in the maritime cases and 0.64 km$^{-1}$ in the continental cases (Fig. 10b). The similarity between the theoretical $\varepsilon_{TKE}$ and the model-diagnosed $\varepsilon_{SC95}$ suggest that the TKE theory adequately captures the simulated trends.

A closer look at CAPE, $z_{cld}$, and $m_b$ again facilitates a physical explanation of the sensitivities of $\varepsilon_{TKE}$ and $\varepsilon_{SC95}$. CAPE and $z_{cld}$ vary significantly between the different experiments, but lack a systematic sensitivity to continentality (Figs. 11a and b). Furthermore, CAPE and $z_{cld}$ compensate each other for all but two experiments when the cloud layer becomes very shallow (Fig. 11c). The $m_b$, on the other hand, is persistently much smaller for the maritime cases than for the continental cases (Fig. 11d). On average, $m_b$ for the continental cases is 2.7 times the $m_b$ value of maritime cases. This trend is owing to stronger sensible heat fluxes over land (Figs. 1a and f), which drive more intense subcloud turbulence. The more vigorous boundary layer updrafts, in turn, generate larger $m_b$ by transporting more kinetic energy across cloud base (Brown et al., 2002, Fig. 11d). When raised to the 2/3 power in Eq. (8), this trend in $m_b$ almost fully explains the sensitivity of cloud dilution to continentality.

The impact of $m_b$ on entrainment can be interpreted using Eq. (7). Disregarding the common $(m_b/w^*)^{1/2}$ factor on both sides, the entrainment term on the left-hand side exhibits a stronger sensitivity to $m_b$ ($\sim m_b^{5/3}$) than does the buoyancy flux/dissipation term on the right-hand side ($\sim m_b^1$). For a given $z_{cld}$, dissipation depends on the turbulent velocity scale $w^*$, which is only a weak function of $m_b$ in Eq. (6). While entrainment also depends on $w^*$, it additionally depends directly on $m_b$, as $\varepsilon m_b$ represents the flux of entrained air into the cloud. The stronger sensitivity of the entrainment term to $m_b$ implies that, when $m_b$ is varied, $\varepsilon$ must act to partially offset these changes. Thus, to maintain the energetic scaling between TKE source terms (buoyancy flux and dissipation) and entrainment flux as $m_b$ increases, $\varepsilon$ must decrease.

To further explore the sensitivity to surface heating, we conduct additional sensitivity experiments with modified sensible heat fluxes, based around the CTRL ARM-SGP case. These additional experiments are conducted without background wind to isolate the impact of buoyancy-driven, rather than shear-driven, turbulence on the dilution rate. In the first set of experiments, we vary the Bowen ratio $\beta$ (the ratio of sensible to latent heat fluxes) by 25% and 50% above and below its control values, while keeping $H + LE$ fixed to the CTRL value. For the second set of experiments, we hold the $LE$ fixed to its CTRL value and only change $H$ by 25% and 50% above and below the control values. The results of both sets of experiments present a consistent picture of larger $m_b$ and, consequentially, weaker $\varepsilon$ for increased surface heating. However, the sensitivity of both $m_b$ and $\varepsilon$ to surface hear flux changes in the $\beta$ experiments were found to be weaker than in the HFX experiments (not shown). This can

be explained by a stronger compensation of the corresponding variations in subcloud turbulence intensity by variations in the subcloud humidity. For example, when $\beta$ is decreased, weaker subcloud turbulence, and hence reduced vertical displacement, is, in part, compensated by increased subcloud humidity, which reduces the amount of vertical displacement required to reach the LFC.

For the HFX experiments, decreasing $H$ tends to reduce the subcloud turbulence while increasing the subcloud specific humidity, with an attendant lowering of the cloud-base. These effects, which are shown for the case with a 50% reduction in $H$ (RHFX50) in Figs. 12j and h, are not unlike those arising from decreased $\beta$. However, their cancellation is weaker—the changes in turbulence intensity dominate. Weaker subcloud updrafts are less able to breach the LFC, leading to decreased $m_{\mathrm{b}}$ and, in turn, increased $\varepsilon$ (Fig. 13). Hence, these findings suggest that it is primarily $H$ that explains the sensitivity to continentality, which is captured by the $m_{\mathrm{b}}$ sensitivity in the TKE scaling.

## 4 Discussion

As discussed in Sect. 1, various studies have found strong correlations between horizontal cloud area and cloud dilution, arguing that wider clouds better protect their inner cores from the suppressive effects of entrainment (Khairoutdinov and Randall, 2006; Kirshbaum and Grant, 2012; Rieck et al., 2014; Rousseau-Rizzi et al., 2017). In the present study, however, this perspective has not yet been taken—only the reverse problem (the impacts of mixing on cloud horizontal area) was considered in Sect. 3.1. In this section, we evaluate the link between cloud width and cloud dilution in more detail.

Satellite observations (e.g., Zhao and Di Girolamo, 2007) and LES studies (e.g., Neggers et al., 2003b) have shown that in shallow-cumulus cloud fields, the vast majority of clouds are small, and larger clouds are few and far-between. The cloud-size distribution has been variously characterized by lognormal, exponential, or power-law functions (e.g., Neggers et al., 2019). The $R_{\mathrm{eff}}$ distributions at the cloud-layer midpoints for the layer-depth sensitivity experiments of Sect. 3.2.1 and Sect. 3.2.2 reveal a similar pattern, with many small and few larger cumuli, broadly resembling lognormal functions (Fig. 14a and b). Because these distributions are similarly shaped, their arithmetic means should provide an adequate reflection of their statistical differences. For the cloud-layer depth sensitivity experiments, the distributions are nearly identical, and so are their arithmetic means (Fig. 14c). In contrast, the subcloud-layer depth sensitivity experiments exhibit a slight shift in the $R_{\mathrm{eff}}$ distribution toward larger values, which is again reflected in the mean profiles (Fig. 14d).

In the cloud-layer-depth sensitivity experiments of Sect. 3.2.1, increased cloud-layer depth led to deeper clouds (as expected) but not to increased cloud widths, at least over the bulk of the cloud layer (Fig. 14c). Thus, increased cloud depth does not always correspond to increased cloud width, which reflects the anisotropic nature of turbulence in conditionally unstable layers (where the saturated updrafts are statically unstable and the unsaturated downdrafts are statically stable). Nevertheless, the similar insensitivity of cloud width and cloud dilution to cloud-layer depth may reflect a causal link between these two metrics.

The larger number of wider clouds for the deeper subcloud-layer depth experiments (Fig. 14d), is reflected by larger $R_{\mathrm{eff}}$ when averaged using the arithmetic mean. A modest sensitivity of cloud width to subcloud-layer depth was found in Sect. 3.2.2, with $R_{\mathrm{eff}}$ at cloud base increasing by around 10 % when the subcloud layer depth increases by 50 % for SCL-DEEP1. This

result suggests that turbulence in the subcloud layer is also anisotropic (though less so than in the cloud layer) and/or that the widest thermals in deeper layers may be suppressed by stronger adverse vertical pressure gradients (e.g., Tang and Kirshbaum, 2020) and therefore exhibit less penetration into the cloud layer. However, the latter speculation is contradicted by the shallow-cumulus observations of Lamer and Kollias (2015), where wider subcloud turbulent structures were found to be associated with stronger updrafts. In any case, the decreased cloud dilution in simulations with wider clouds again is consistent with the notion of wider clouds being less diluted.

The cloud-layer RH experiments, however, reveal that cloud size and dilution rate are not always negatively correlated. The clouds that form in the drier environments exhibit smaller $R_{\mathrm{eff}}$ but are *less* diluted than those in moister environments (Figs. 4 and 15). In these cases, the smaller dilution rates in the drier flows are caused by the detrainment of negatively buoyant mixtures of entrained air, and this detrainment leads to a narrowing of the cloud. Thus, while cloud size may play an important role in regulating cloud dilution, it is also a reflection of the dilution and detrainment history of the cloud.

Similarly, cloud size alone is not a reliable indicator of the dilution rate for the different maritime and continental cases (Fig. 16). While the larger clouds in ARM-SGP are less diluted than the shallower clouds in BOMEX, clouds of similar if not larger size in RICO are even more diluted than those in BOMEX (Fig. 10a). The $\varepsilon_{\mathrm{SC95}}$ in RICO is the largest of all cases under consideration while the $R_{\mathrm{eff}}$ in RICO is comparable to some RACORO cases (Fig. 16). The cloud-size variation of the different RACORO cases (each of the three days is counted as a separate case) is due to the changes in the large-scale forcings. These findings reinforce that the horizontal cloud size does not single-handedly explain all of the sensitivities of cloud dilution. It is important to note that CTRL-HR simulations exhibit systematically smaller clouds than the experiments with lower grid spacings (Hanley et al., 2015).

A more robust physical interpretation of the majority of the cloud-dilution sensitivities evaluated herein is provided by the TKE similarity theory. This theory has been used to gain insight into the weak sensitivity to the cloud-layer depth, the weak dependence on subcloud-layer depth, and the strong sensitivity to continentality. The importance of $m_{\mathrm{b}}$ on dilution reflects the broader importance of subcloud dynamics on cloud-layer convection, a crucial link that is being increasingly recognized (e.g., Tang and Kirshbaum, 2020). Both the initial cloud properties near cloud base (nature) and the environmental conditions experienced by the cloud as it ascends (nurture) have been examined for their roles in cloud evolution (e.g., Dawe and Austin, 2012; Romps and Kuang, 2010; Rousseau-Rizzi et al., 2017). Defining nature as the thermodynamic and kinematic state of a cloudy parcel at cloud base, Romps and Kuang (2010) analysed the relative importance of nature vs nurture from a parcel perspective. In LESs of shallow cumuli, they found only a very weak correlation between the parcel's cloud-layer properties and its initial conditions at cloud base. They thus concluded that nature is of secondary importance for cloud evolution. In contrast, Dawe and Austin (2012) considered thermodynamic conditions as well as morphological characteristics of whole cloud entities as nature. While nurture primarily regulated the cloud thermodynamic properties, nature played an important role in controlling the cloud width and height in the upper cloud layer. Our results are consistent with Dawe and Austin (2012) in that cloud-base conditions may leave an imprint on the cloud properties above.

The TKE theory was not used to explain the moderate sensitivity to cloud-layer RH because environmental humidity is neglected in the theoretical formulation (Grant and Brown, 1999; Grant and Lock, 2004). Moreover, it cannot explain sensitiv-

ities to the vertical wind profile, which, as will be shown in a companion paper, can also strongly impact cloud dilution and detrainment.

The buoyancy-sorting concept provides a useful physical explanation of the sensitivity of cloud dilution to cloud-layer RH found herein, as well as that in the numerical simulations of Stirling and Stratton (2012) and Bera and Prabha (2019) and the observations of Lu et al. (2018). For a given amount of cloud-environmental mixing, a smaller fraction of entrained air becomes negatively buoyant and detrains in moister environments, resulting in more diluted cloud cores. On the other hand, our findings contrast with the conclusion of Derbyshire et al. (2004) and de Rooy et al. (2013) that cloud dilution decreases with increasing cloud-layer RH. However, their simulations explored a very different part of parameter space, with much larger cloud-layer RH variations (25 to 90 %) and much deeper, precipitating convection.

The limited range of cloud-layer RH sampled our experiments stemmed from the need to maintain a certain level of consistency between different sensitivity tests. To generate comparable cloud fields in the different experiments, we held the cloud-base RH fixed and only modified the cloud-top RH (with a linear variation of $q_v$ in-between). As a result, even large changes in the latter did not greatly change the layer-averaged RH. Therefore, we caution that the positive sensitivity of shallow-cumulus dilution to the small changes in layer-averaged RH found herein may not apply throughout the very large parameter space of atmospheric convection.

Although simulated dilution rates for shallow cumuli tend to be larger than those for deep cumuli (e.g., Khairoutdinov and Randall, 2006; Del Genio and Wu, 2010), the sensitivity of the dilution rate to cloud-layer depth within the shallow cumulus regime was found to be minimal, at least for a fixed cloud-layer lapse rate. Similar to the cloud-layer RH experiments, the small range of cloud-layer depth from 1.1 in CL-SHAL1 to 1.8 km in CL-DEEP2 limits the generality of the result. However, extending the experiments to even deeper cloud layers is problematic since the characteristics of the cloud field would change significantly. In particular, deeper clouds would begin to produce substantial precipitation, which could change their morphology and environmental sensitivities (Khairoutdinov and Randall, 2006; Grant, 2007). Similar considerations also explain why the subcloud-layer depth was not changed more substantially in the corresponding experiments.

Neggers et al. (2002) developed a multiparcel entrainment model for shallow cumulus convection, in which dilution was prescribed to be inversely proportional to the vertical velocity ($w$). The reasoning behind this sensitivity is that, for a faster ascending air parcel, entrainment has less time to dilute the cloudy parcel than for a slower rising one. Subsequent studies have supported these findings and formulated more complex relationships between core properties and $\varepsilon$. Tan et al. (2018), for example, parameterized $\varepsilon$ using a combination of cloud buoyancy ($b$) and $w$:

$$\varepsilon_{\text{Tan}} = c_\varepsilon \frac{\max(0, b)}{w^2} . \tag{10}$$

We calculate $\varepsilon_{\text{Tan}}$ using bulk core statistics and compare it to the calculated $\varepsilon_{\text{SC95}}$. With the coefficient $c_\varepsilon$ set to 0.3 (instead of 0.12 as suggested by Tan et al. (2018)), $\varepsilon_{\text{Tan}}$ captures the overall trend of larger $\varepsilon$ in maritime clouds and smaller $\varepsilon$ in continental clouds (Fig. 17). However, this relation cannot explain all of the sensitivities found in the experiments. For example, the slightly larger $\varepsilon$ in RICO, relative to BOMEX, is not captured, and the differences between ARM-SGP and RACORO are

485 over-predicted.. Thus, additional factors beyond $b$ and $w$ may be required to more accurately represent the sensitivity of cloud dilution to environmental conditions.

## 5 Conclusions

This study has advanced the physical understanding of cumulus dilution and its sensitivity to environmental conditions using large-eddy simulation. As used herein, the term "dilution" is synonymous to bulk entrainment, where cloud-environmental
mixing is inferred through the vertical gradient of conditionally averaged variables within the buoyant cloud core. The term "entrainment" is reserved for all of the air ingested into the cloud, even if the air differs from environmental air and/or does not contribute to cloud-core dilution. For generality and to explore the role of continentality, both maritime and continental shallow-cumulus cloud ensembles were simulated. The sensitivity of cloud dilution to four different environmental thermodynamic parameters, including cloud-layer relative humidity (RH), cloud- and subcloud-layer depths, and continentality, was quantified
and interpreted. These conditions were chosen because the sensitivities of dilution to them are either uncertain (cloud-layer RH), unexplained (continentality), and/or largely unexamined (cloud- and subcloud-layer depths).

  Systematic experiments with different initial thermodynamic profiles revealed a positive correlation between cumulus dilution and cloud-layer RH. Specifically, a 10 %-increase in RH led to a 25 % and 47 % increase in fractional dilution in the maritime BOMEX case and the continental ARM-SGP case, respectively. This finding was explained by the concept of buoy-
500 ancy sorting (Fig. 4): in drier environments, a larger fraction of entrained air becomes negatively buoyant and detrains, leaving a smaller but less dilute cloud core. This trend is consistent with the observations of Lu et al. (2018) and the simulations of Stirling and Stratton (2012) and Bera and Prabha (2019), but inconsistent with Derbyshire et al. (2004) and de Rooy et al. (2013) as well as the prescribed sensitivity of entrainment to RH in the Bechtold et al. (2008) cumulus parameterization.

  Moreover, cloud dilution was found to be strongly sensitive to continentality, with fractional dilution rates in maritime clouds
about twice those in continental clouds. To explain this sensitivity, a similarity theory of shallow-cumulus transports based on the turbulent-kinetic-energy (TKE) budget was used (Grant and Brown, 1999; Grant and Lock, 2004; Kirshbaum and Grant, 2012). This theory suggests that this sensitivity can be attributed to a larger cloud-base mass flux ($m_b$) over land, driven by larger surface sensible heat fluxes and more intense subcloud turbulence. From an energetics perspective, this sensitivity arises from the theoretical assumption that the kinetic energy imparted to entrained air scales with the dominant TKE source terms
(buoyancy flux and dissipation). The former is more sensitive to the cloud-base mass flux than the latter, and thus, to maintain equilibrium, changes in $m_b$ must be partially offset by changes in the cloud dilution rate.

  Relative to the sensitivities of cloud dilution to cloud-layer RH and continentality, the sensitivities to both cloud-layer depth (a 3 % change in dilution for a doubling of the layer depth) and subcloud-layer depth (a 4 % decrease in dilution for an increase of the layer depth by 50 %) were weak. These minimal sensitivities may be surprising given the tendency for simulated cloud
dilution to weaken during the transition from shallow to deep convection (e.g., Khairoutdinov and Randall, 2006; Del Genio and Wu, 2010). Using the TKE similarity theory, the weak sensitivity to cloud-layer depth was interpreted to result from the largely offsetting effects of increased moist instability, which tends to increase dilution, and increased cloud depth, which

tends to weaken dilution. The slightly stronger sensitivity of cloud dilution to subcloud-layer depth, like the sensitivity to continentality, was dominated by changes in cloud-base mass flux. Deeper subcloud layers produced larger cloud-base mass fluxes, which thereby decreased the dilution rate.

The above physical interpretations do not invoke a causal relationship between horizontal cloud size and cloud dilution, which is a common perspective for explaining entrainment/dilution sensitivities (e.g., Khairoutdinov and Randall, 2006; Kirshbaum and Grant, 2012; Rousseau-Rizzi et al., 2017). In particular, it is often found that wider clouds tend to undergo less dilution than narrower clouds. The cloud-size perspective was found to qualitatively apply to the cloud-layer and subcloud-layer depth experiments, where minimal to slight increases cloud size were accompanied by minimal to slight decreases in dilution. However, this perspective did not apply to the cloud-layer RH or continentality experiments, where the relationship between cloud size and dilution was reversed or inconclusive. Based on these findings, we support the conclusion of Hannah (2017) that the relationship between cloud size and cloud dilution is complex and interactive, and inferences concerning this relationship should be made with care.

Additional steps are needed to strengthen the understanding of shallow-cumulus dilution and improve parameterizations of shallow convection in large-scale models, most of which do not explicitly account for the sensitivities found herein (e.g., Webb et al., 2015; Lu et al., 2016). One important step is to identify and examine the full range of environmental sensitivities of cloud dilution, which may extend beyond the parameters considered to date. As a step in this direction, a companion paper will examine, using a similar LES approach, the impacts of the vertical wind profile on cloud dilution. However, perhaps the most efficient way to identify all of the sensitivities of cloud dilution is through climatological observations of cloud dilution in different locations. Such an ambitious objective may be possible using one or more of the ground-based dilution retrievals evaluated in Drueke et al. (2019), and future efforts are planned to conduct long-term retrievals at multiple ARM observatories.

For brevity, the focus of this study was placed on the sensitivity of cloud dilution to environmental conditions. However, since entrainment and dilution relate primarily to the inflow of surrounding air into the cloud, their counterpart—cloud outflow and detrainment—demand further analysis. de Rooy and Siebesma (2008) found detrainment to be sensitive to two environmental factors: cloud-layer depth and relative humidity. In more humid environments, entrainment of environmental air leads to less evaporative cooling, less buoyancy reversal, and hence less detrainment. Our cloud-layer RH results agree well with this finding. Nevertheless, a more complete study of the sensitivity of detrainment to environmental conditions remains outstanding and is deferred to future work.

*Code and data availability.*  The Bryan Cloud Model (CM1) is available under http://www2.mmm.ucar.edu/people/bryan/cm1/. Primary data and scripts used in the analysis and other supplementary information that may be useful for reproducing the author's work are archived by the Department of Atmospheric and Oceanic Sciences (McGill University) and can be obtained by contacting sonja.drueke@mail.mcgill.ca.

*Author contributions.* S.D. and D.J.K. developed the scientific question, and S.D. conducted the simulations and carried out the analysis under the supervision of D.J.K. and co-supervision of P.K. S.D. prepared the paper with contributions from D.J.K. and P.K.

*Competing interests.* The authors declare that they have no conflict of interest.

*Acknowledgements.* Research funding was provided from the Natural Sciences and Engineering Research Council (NSERC) Grant NSERC/RGPIN 418372-17 and the US Department of Energy Atmospheric System Research (DOE–ASR) Program under contract DE-SC0020083. P.K.'s contributions were supported by DOE–ASR under contract DE-SC0012704. The numerical simulations were performed on the Guillimin supercomputer at McGill University and Béluga supercomputer at the École de technologie supérieure, both under the auspices of Calcul

Québec and Compute Canada. We thank Satoshi Endo, Phil Austin, and George Bryan for their valuable contributions.

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

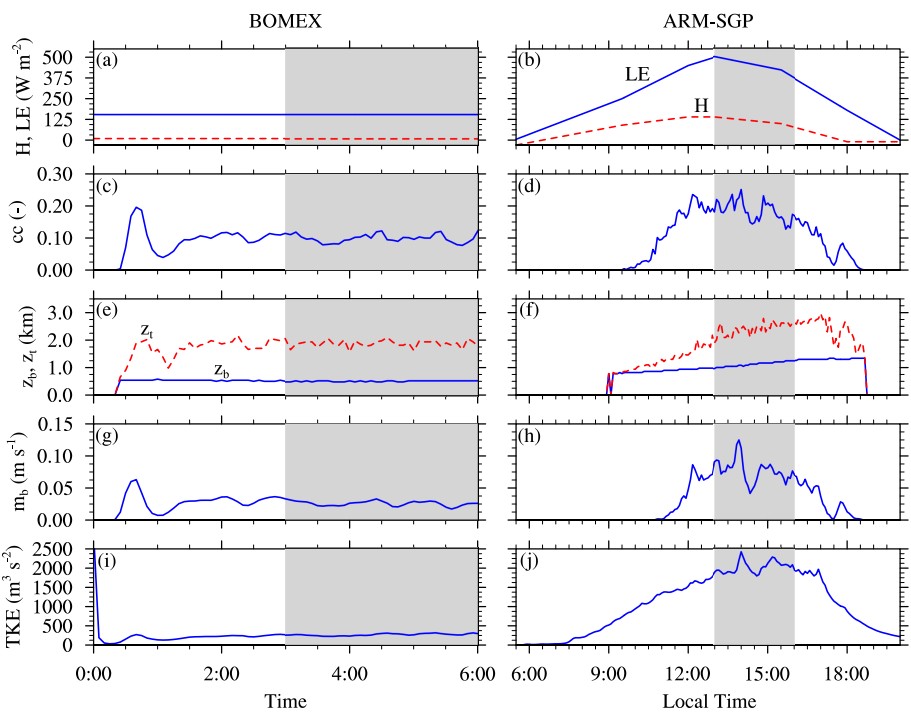

**Figure 1.** Time series of (a-b) latent and sensible heat fluxes, (c-d) total cloud cover, (e-f) cloud base and cloud top height, (g-h) cloud core base mass flux, and (i-j) vertically integrated TKE. The first column shows the time series for the BOMEX CTRL simulations, and the second column depicts the temporal evolution of the ARM-SGP CTRL. The gray shading indicates the time window over which averages have been performed.

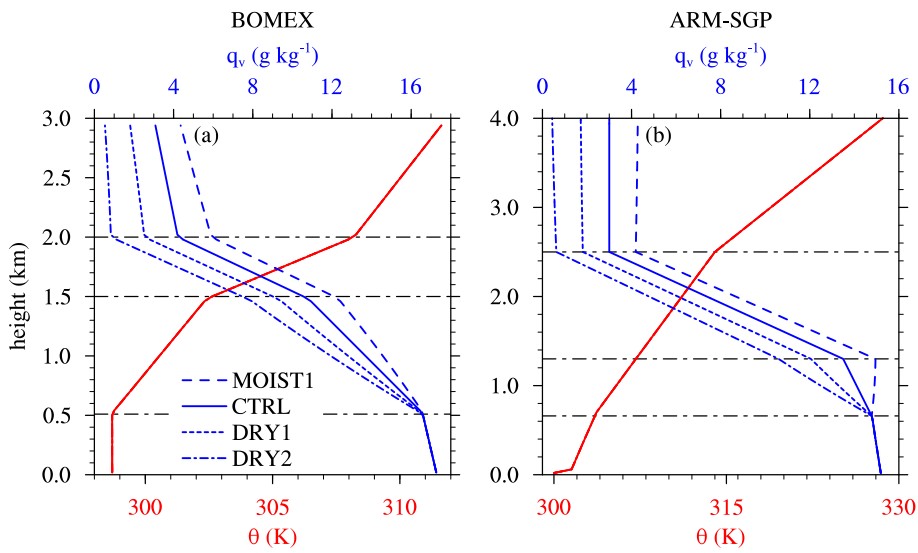

**Figure 2.** Initial profiles of potential temperature ($\theta$) and water vapor mixing ratio ($q_v$) for the cloud-layer RH experiments for (a) BOMEX and (b) ARM-SGP.

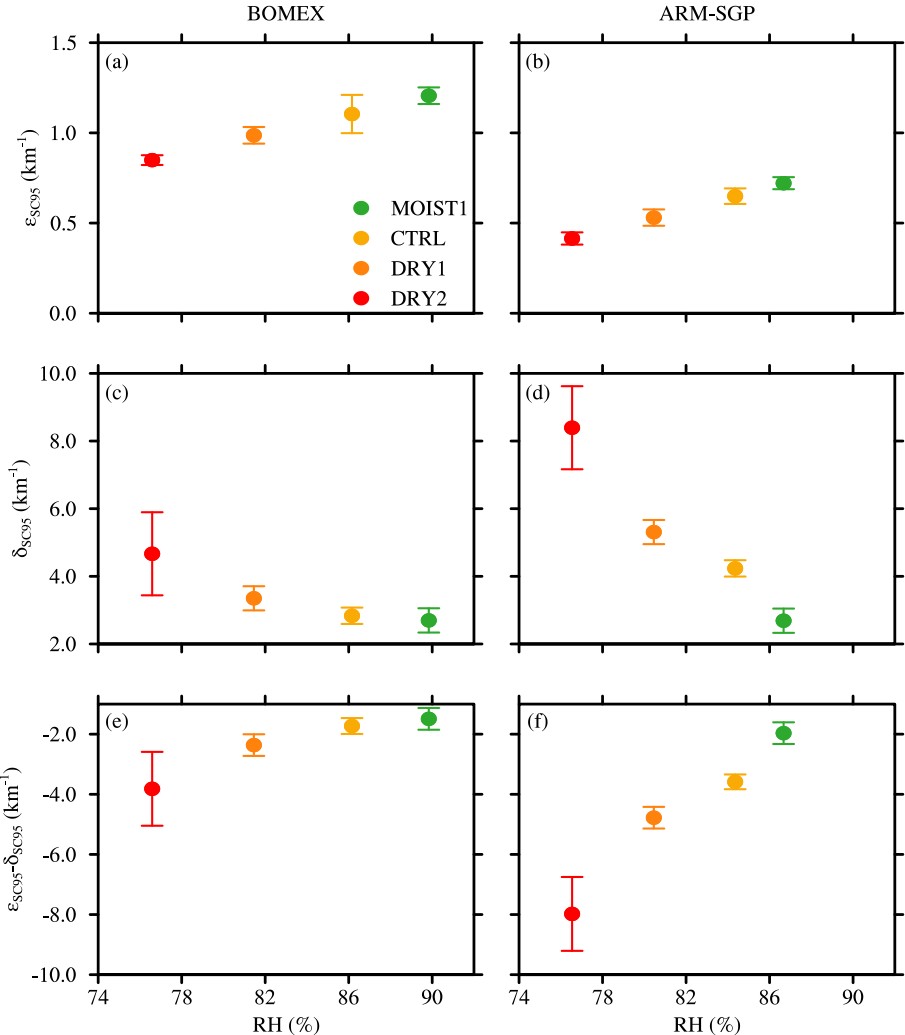

**Figure 3.** Variation of mean (a-b) fractional dilution rate ($\varepsilon$), (c-d) fractional detrainment rate ($\delta$), and (e-f) the difference of fractional dilution and detrainment ($\varepsilon - \delta$) as a function of cloud-layer RH (vertically averaged over the middle 50 % of the cloud layer). The left column shows the maritime BOMEX case and the right column shows the continental ARM-SGP case. The error bars indicate the width of twice the standard deviation of the ensembles.

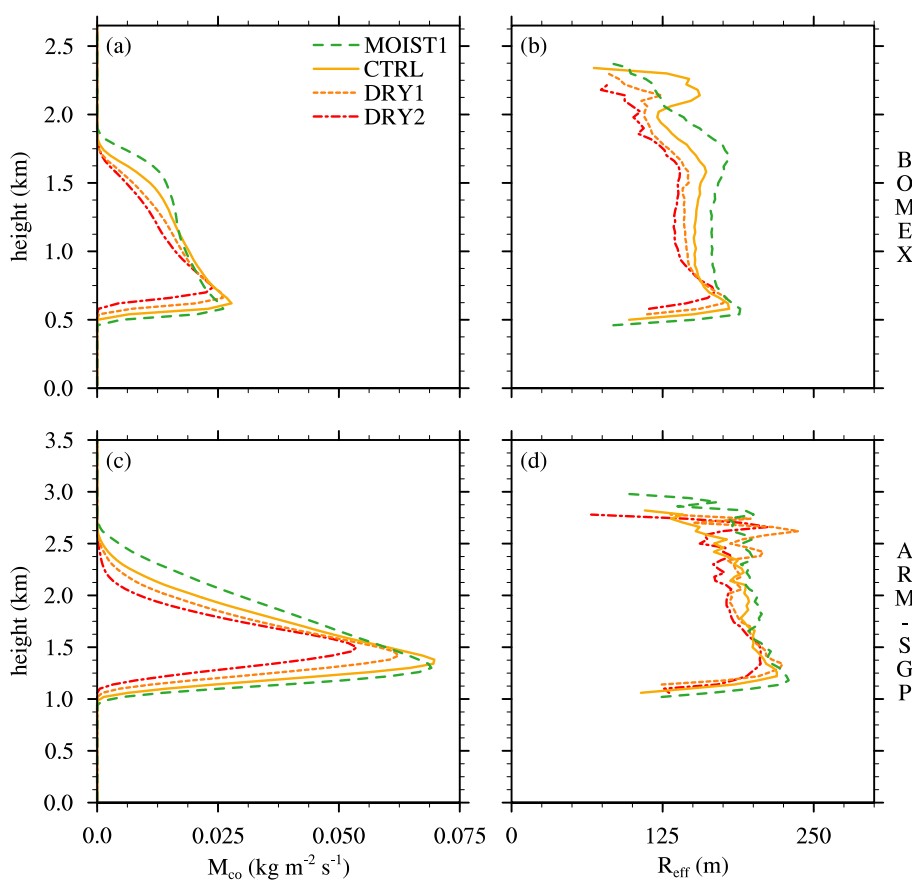

**Figure 4.** Conditionally averaged cloud-core mass flux ($M_{co}$) and effective cloud radius ($R_{eff}$) for the cloud-layer RH experiments for (a-b) BOMEX and (c-d) ARM-SGP, respectively.

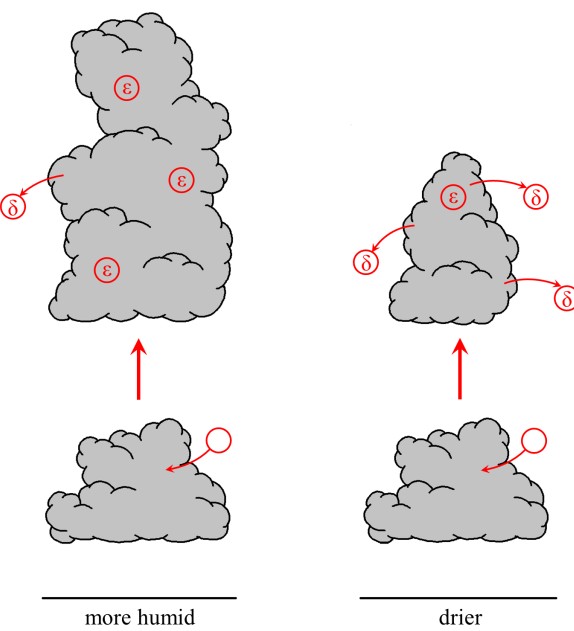

**Figure 5.** Schematic of cloud development in more (left) and less (right) humid environments. In drier environments, a larger fraction of entrained air becomes negatively buoyant and detrains, leaving a smaller cloud.

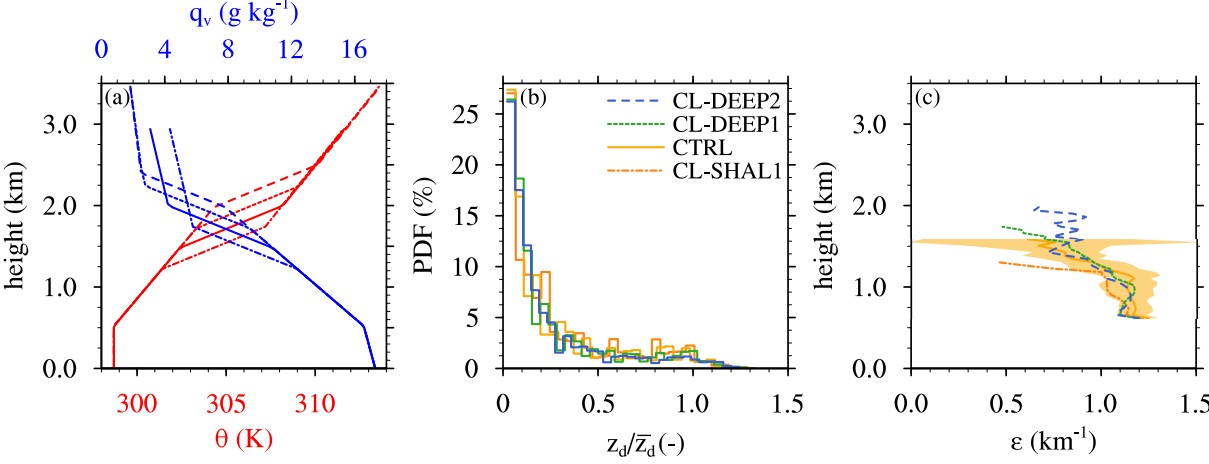

**Figure 6.** (a) Initial profiles of potential temperature ($\theta$) and water vapor mixing ratio ($q_v$) for the experiments with different cloud-layer depths. (b) The probability density function (PDF) of individual cloud-top heights as a fraction of the cloud-layer top and (c) the fractional-dilution-rate profiles for the cloud-layer-depth experiments.

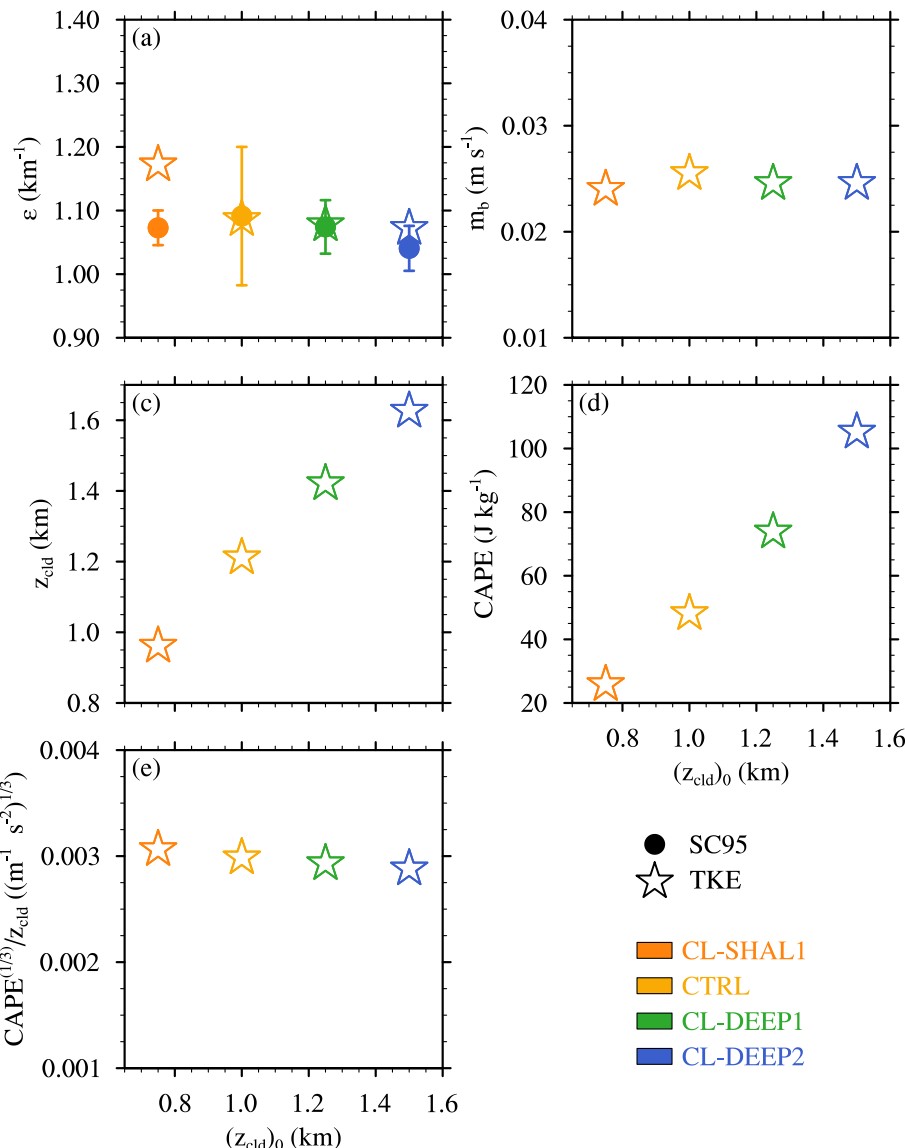

**Figure 7.** Analysis of experiments with different initial cloud-layer depths ($(z_{cld})_0$): (a) fractional dilution rate ($\varepsilon$) calculated using Eq. (1) and estimated using Eq. (8), (b) cloud-base mass flux ($m_b$), (c) cloud-layer depth ($z_{cld}$), (d) convective potential available energy (CAPE), and (e) $CAPE^{(1/3)}/z_{cld}$. The error bars in (a) indicate the width of twice the standard deviation of $\varepsilon_{SC95}$.

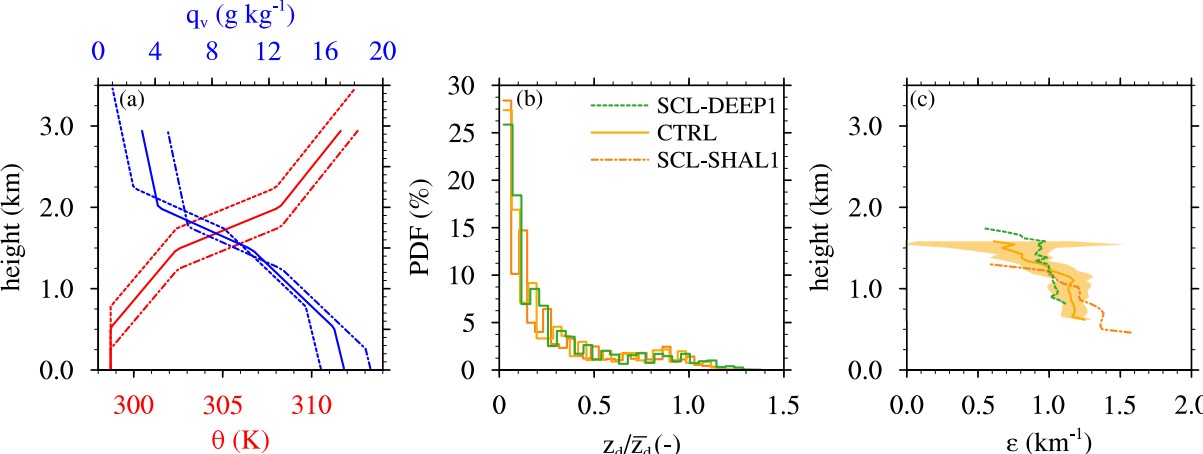

**Figure 8.** Same as Fig. 6 but for subcloud-layer depth experiments.

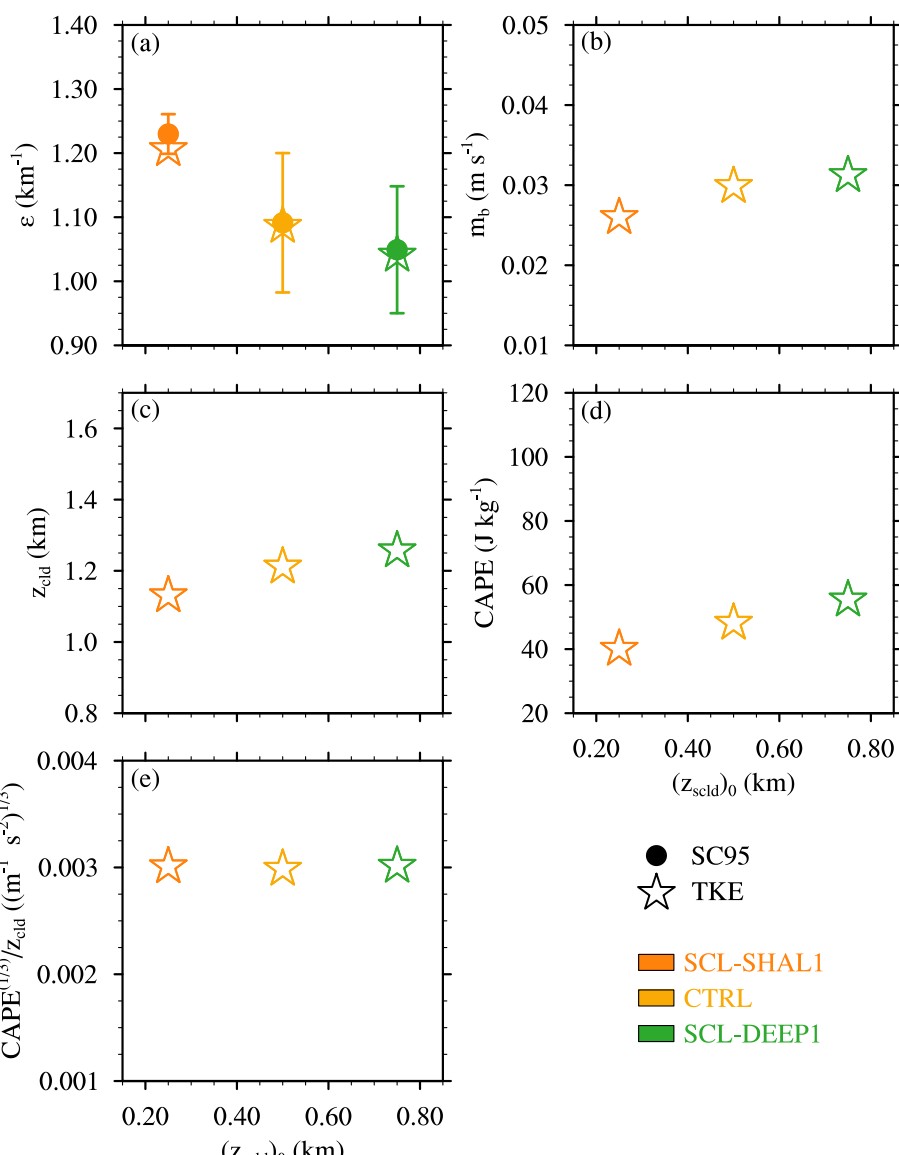

**Figure 9.** Same as Fig. 7 but for subcloud-layer depth experiments.

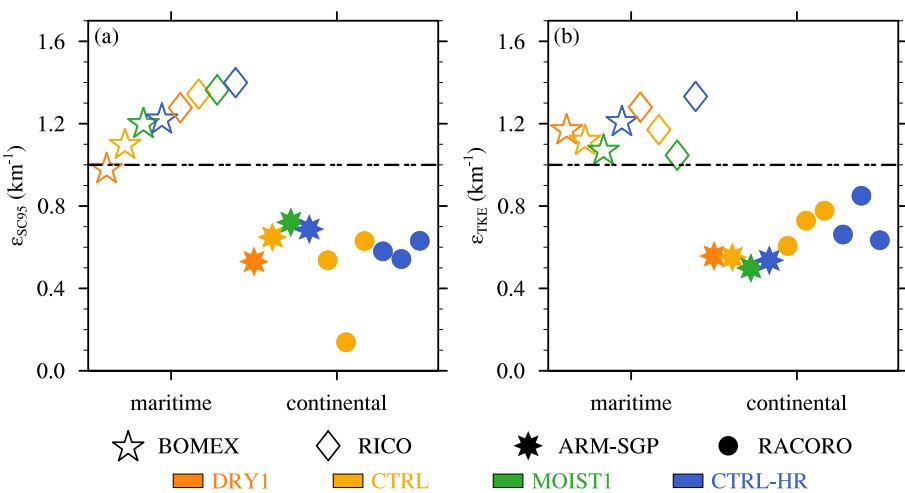

**Figure 10.** Fractional dilution rates for the 18 different cases (DRY1, CTRL, MOIST1, and CTRL-HR for BOMEX, RICO, and ARM-SGP and CTRL and CTRL-HR for RACORO (each of the three days counts as a separate case)) for maritime cases (BOMEX and RICO) and continental cases (ARM-SGP and RACORO) calculated using (a) the SC95 diagnostic and (b) TKE estimate.

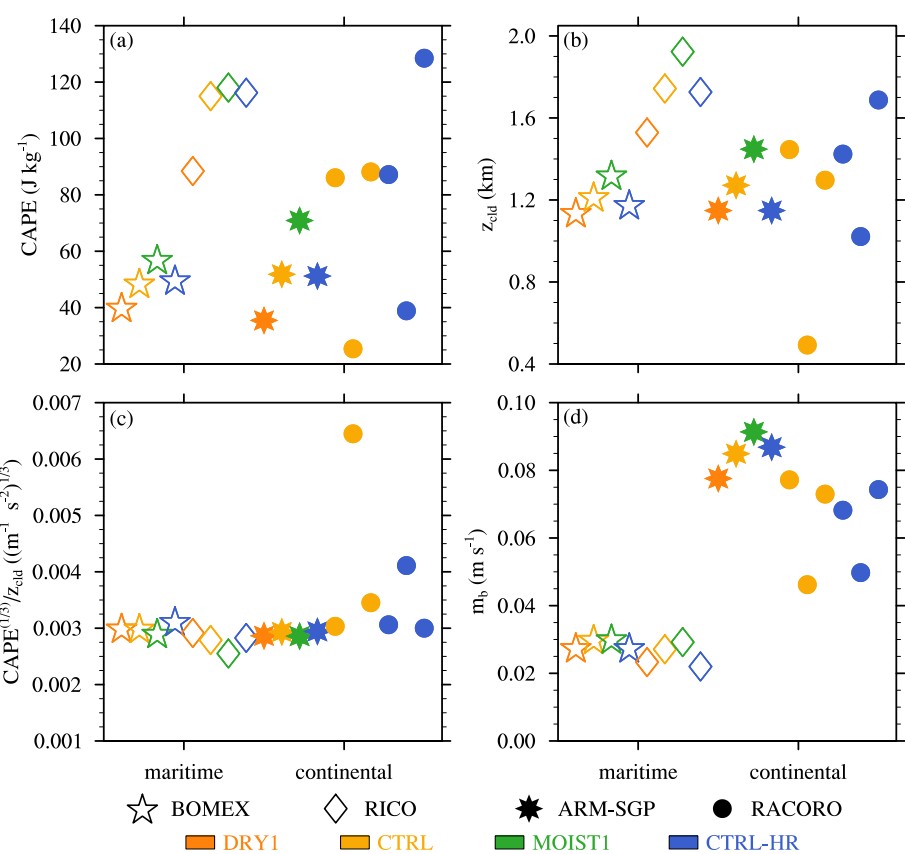

**Figure 11.** Analysis of the maritime and continental experiments: (a) convective potential available energy (CAPE), (b) cloud-layer depth ($z_{cld}$), (c) $CAPE^{(1/3)}/z_{cld}$, and (d) cloud-base mass flux ($m_b$).

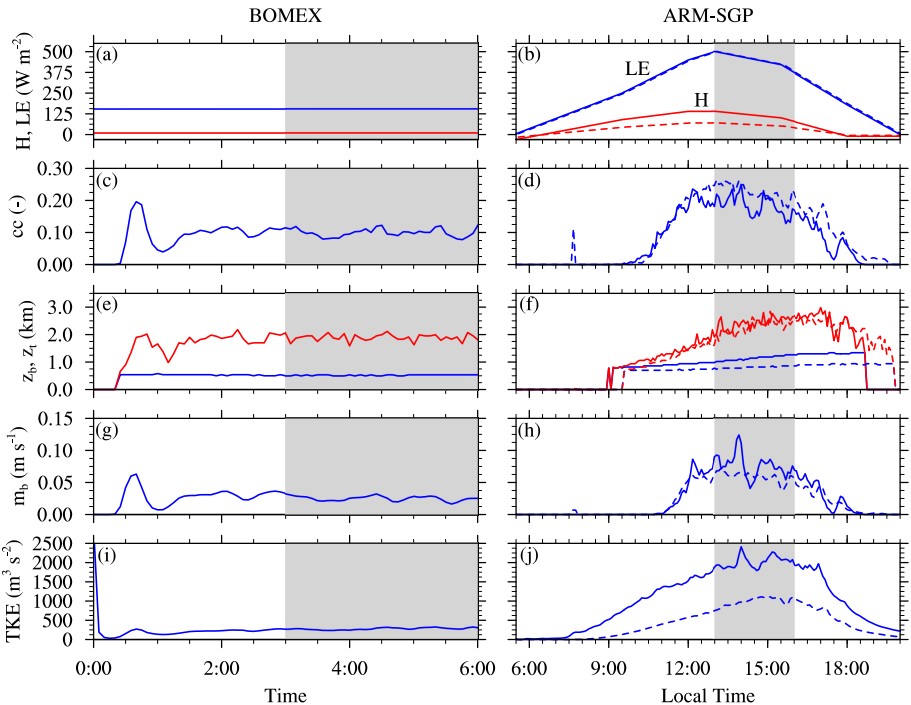

**Figure 12.** Same as Fig. 1 but in addition the time series of the diurnal cycle of the ARM-SGP experiments with a reduced $H$ by 50% (RHFX50) are included in the dashed lines in the second column. The gray shading indicates the time window over which averages have been performed.

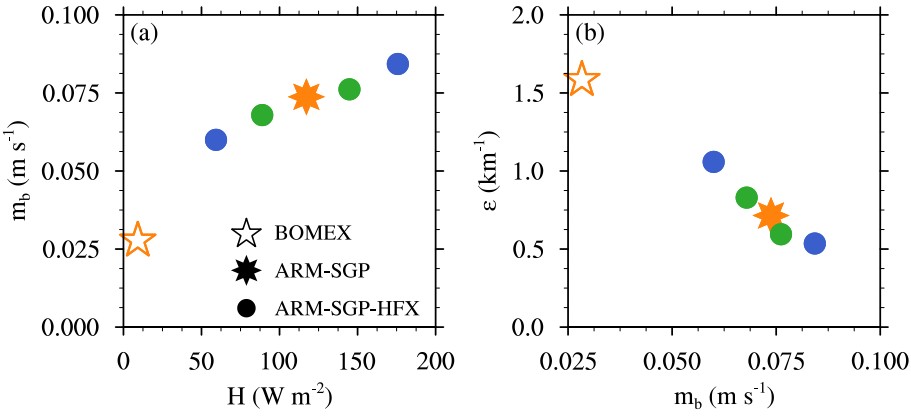

**Figure 13.** (a) The cloud-base mass flux ($m_b$) as a function of sensible heat flux ($H$) and (b) the dilution rate ($\varepsilon$) averaged over the central 50% of the cloud layer as a function of $m_b$ for BOMEX and ARM-SGP as well as the ARM-SGP-HFX experiments.

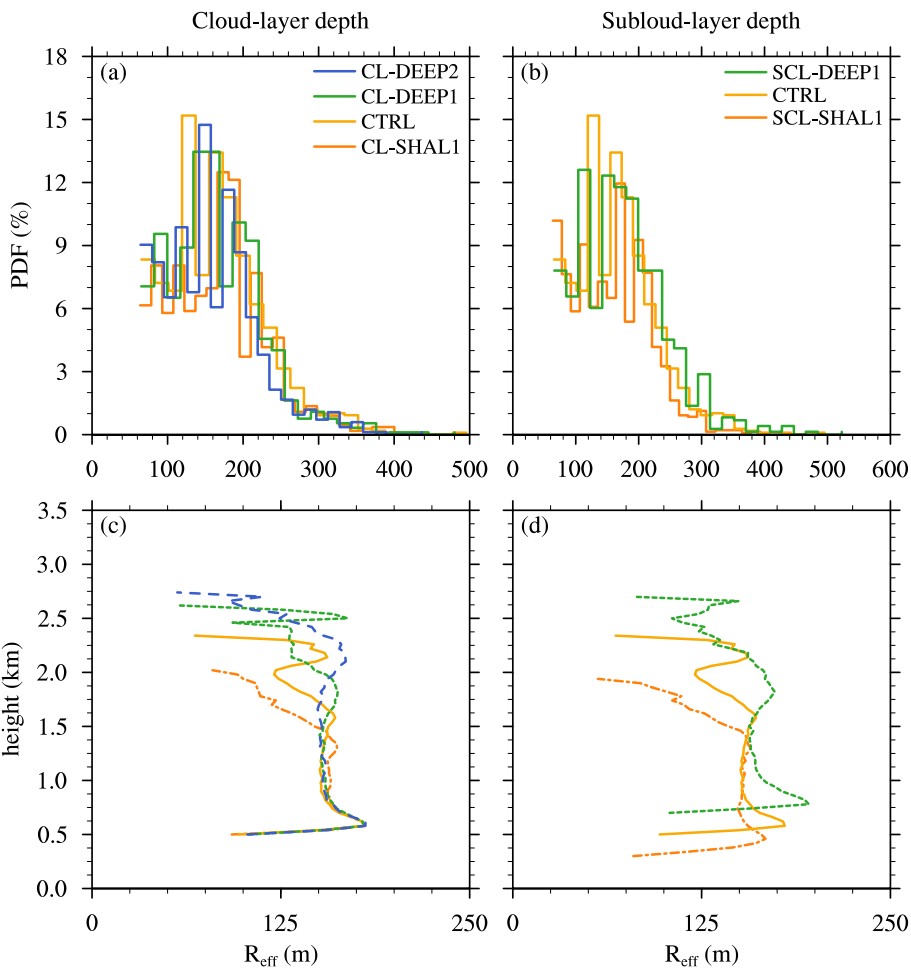

**Figure 14.** PDF of the effective cloud radius ($R_{\text{eff}}$) halfway into the the respective cloud layers for the experiments with (a) different cloud-layer depths and (b) varying subcloud-layer depths. The averaged $R_{\text{eff}}$ profile for the same experiments are shown in (c) and (d), respectively.

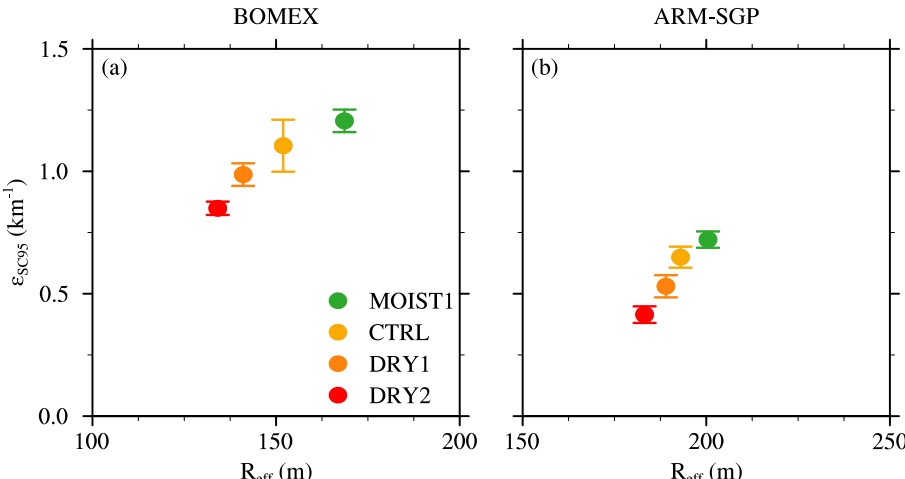

**Figure 15.** Fractional dilution rate ($\varepsilon_{SC95}$) as a function of effective cloud radius ($R_{eff}$) for (a) the maritime BOMEX cases and (b) the continental ARM-SGP cases.

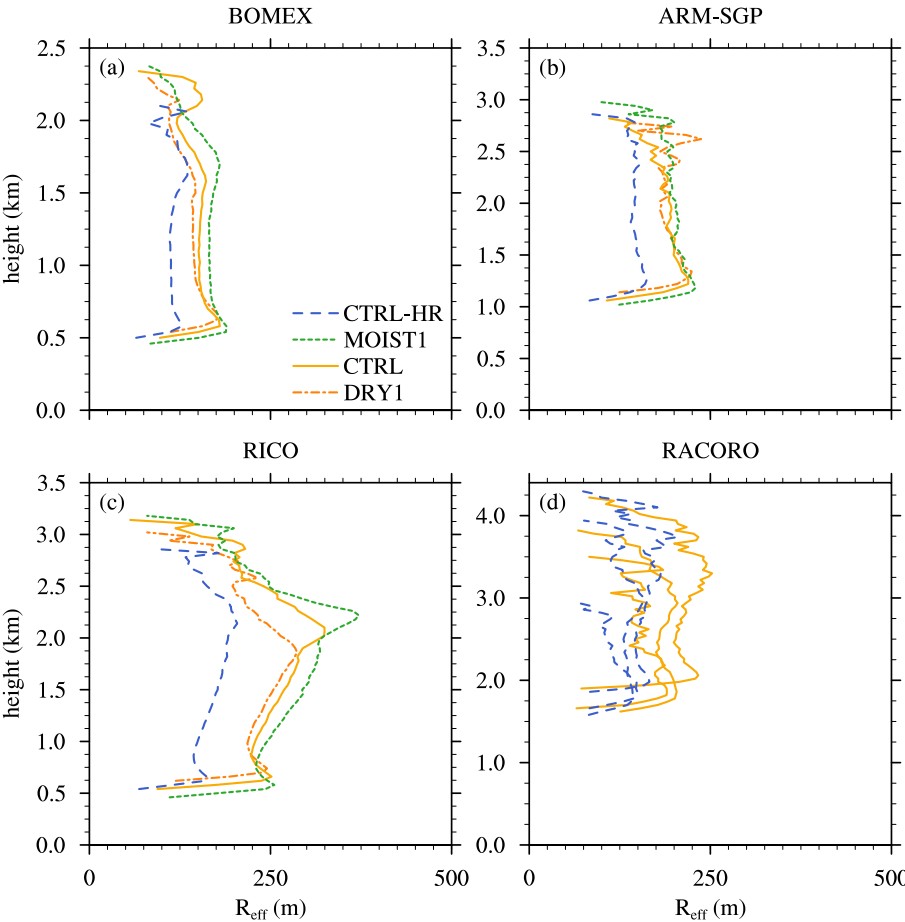

**Figure 16.** The effective cloud radius ($R_{\text{eff}}$) for maritime and continental experiments: (a) BOMEX, (b) ARM-SGP, (c) RICO, and (d) RACORO. For RACORO each of the three days counts as a separate case.

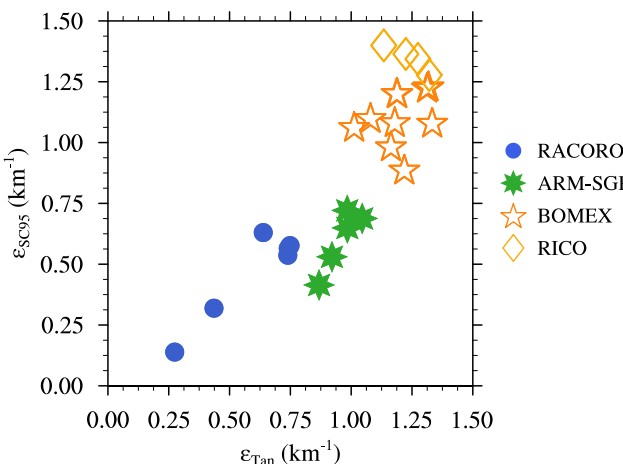

**Figure 17.** Relationship between dilution rate calculated using Eq. (1) ($\varepsilon_{SC95}$) and dilution rates approximated using Eq. (26) in Tan et al. (2018) for all 18 experiments. The non-dimensional coefficients $c_\varepsilon$ has been set to 0.3.

**Table 1.** Numerical configurations for the different experiments. $\Delta_\mathrm{x}$, $\Delta_\mathrm{y}$, and $\Delta_\mathrm{z}$ are the $x$, $y$, and $z$ grid spacings, respectively, $L$ is the horizontal domain size, $D$ is the domain height, $T$ is the model integration time, $W$ is the large-scale subsidence, $(\partial q_\mathrm{v}/\partial t)_\mathrm{adv}$ is the large-scale horizontal advection of moisture, $(\partial \theta/\partial t)_\mathrm{adv}$ is the large-scale horizontal advection of heat, $Q_\mathrm{r}$ is the radiative cooling rate, $H$ is the sensible heat and $LE$ the latent heat flux, $c_\mathrm{d}$ is the surface drag, and $N_\mathrm{CCN}$ is the droplet number concentration. "p" refers to settings that are prescribed based on cited publications. S. Endo (personal communication) has provide the large-scale forcing and surface fluxes for RACORO. The (*) indicates that the values or time series are determined interactively according to formulae in published work. We refer interested readers to the relevant publications for further details.

| | maritime cases | | continental cases | |
|---|---|---|---|---|
| | BOMEX | RICO | ARM-SGP | RACORO |
| $\Delta_\mathrm{x}$, $\Delta_\mathrm{y}$ (m) | 100 (50) | 100 (50) | 64 (32) | 150 (75) |
| $\Delta_\mathrm{z}$ (m) | 40 | 40 | 40 | 40 |
| $L$ (km$^2$) | 6.4×6.4 | 12.8×12.8 | 6.4×6.4 | 8.0×8.0 |
| $D$ (km) | 3.0 | 4.0 | 4.4 | 5.0 |
| $T$ (h) | 6 | 24 | 14.5 | 60 |
| Latitude (°N) | 14.9 | 18 | 36 | 36 |
| $W$ | p | p | - | p |
| $(\partial q_\mathrm{v}/\partial t)_\mathrm{adv}$ | p | p | p | p |
| $(\partial \theta/\partial t)_\mathrm{adv}$ | - | p | p | p |
| $Q_\mathrm{r}$ | p | p | p | p |
| surface fluxes | p | * | p | p |
| $\quad H$ (W m$^{-2}$) | 10 | 5 – 10 | -30 – 140 | -26 – 161 |
| $\quad LE$ (W m$^{-2}$) | 157 | 130 – 191 | 0 – 500 | -7 – 329 |
| $c_\mathrm{d}$ | * | p | * | * |
| $N_\mathrm{CCN}$ (cm$^{-3}$) | 100 | 70 | 250 | 500 |

**Table 2.** Summary of numerical experiments. See text for further details.

|  | maritime | | continental | |
| --- | --- | --- | --- | --- |
|  | BOMEX | RICO | ARM-SGP | RACORO |
| CTRL | x | x | x | x |
| CTRL HR | x | x | x | x |
| cloud-layer humidity | | | | |
| MOIST1 | x | x | x | - |
| DRY1 | x | x | x | - |
| DRY2 | x | - | x | - |
| cloud-layer depth | | | | |
| CL-SHAL1 | x | - | - | - |
| CL-DEEP1 | x | - | - | - |
| CL-DEEP2 | x | - | - | - |
| subcloud-layer depth | | | | |
| SCL-SHAL1 | x | - | - | - |
| SCL-DEEP1 | x | - | - | - |