# Peer review of "Environmental sensitivities of shallow-cumulus dilution. Part I: Selected thermodynamic conditions"

_Atmospheric Chemistry and Physics, 2020_

## Referee Comment (RC1) · Anonymous Referee #1 · 3 Jun 2020

This paper discusses the sensitivity of dilution of shallow convection as a function of large scale state. While the paper is not unique, it does yield another piece in the puzzle of figuring out how to parameterize entrainment. The paper is generally well written, but sometimes also a little descriptive, with the theoretical interpretation mainly hypothesized.

To enhance the paper, I have the following suggestions. I realize that not all of them may be feasible within the scope of this paper.

1) The main finding to me is the dependence on 'continentality'. This is of course in reality more a dependence on the surface fluxes, and in itself it is no surprise that maritime clouds are different from continental once. So I would enjoy seeing this explored a bit further, for instance by looking into whether this is more an effect of the total buoyancy flux, or of the evaporative fraction/Bowen ratio. IN other words: Is this about the latent heat or the sensible heat?

2) The finding that the cloud base mass flux can explain the difference between the two regimes agrees with Dawe and Austin, and conflicts with Romps' Nature vs Nurture concept. Some discussion of that would help here

3) Detrainment is at least as important for cloud evolution as entrainment is. Is there a reason to barely include detrainment in this paper?

4) The cloud depth discussion is a bit too far simplified, as most clouds in a shallow Cu distribution would not come close to the cloud layer top, and therefore would not "feel" the extended depth of the layer. So what happens if you only sample clouds that actually did make it to the cloud layer top? Is there also some response in other variables here? Think of cloud (core) fraction, fluxes, etc.

5) Similarly, the subcloud layer alteration is a serious disturbance to the flow, and it seems like we are merely looking at the transient here. I am not sure I am learning a lot from that, so I recommend either removing the section, or clarifying its value.

6) For the change in cloud layer humidity, the argument is that changes are based on the relative difference between environment and cloud, and by extension between environment and sub-cloud. If it is the relative difference, does that mean that it is only the gradient that matters? So if one would shift the entire profile, no response would be visible? **ACPD**

---

## Referee Comment (RC2) · Anonymous Referee #2 · 27 Jul 2020

This is interesting, carefully done investigation into the impact of environmental relative humidity, cloud and subcloud layer depth, and surface sensible and latent heat fluxes on mixing and dilutiion in shallow clouds. The use of TKE scaling arguments allow the authors to bring some clarity into the broad range of effects that the environment has on cumulus mixing. Minor comments:

1) Resolution: I wasn't clear what the meaning of the parenthetical 100(50) notation was regarding  $\Delta x$  and  $\Delta y$  in Table 1. Does the 50 indicate that the same run was done at that finer resolution, and the results didn't change? It would be helpful to clarify this, and if possible to confirm that parameters like the cloud base mass flux didn't change

with changing resolution.

2) Cioud size distribution: Satellite observations (e.g. Zhou and Di Girolamo, 2007 doi:10.1029/2006JD007371) indicate that cloud sizes follow a powerlaw distribution, so that the simple arithmetic mean isn't particularly representative of the actual size pdf. Feingold et al. (2017) doi: 10.1002/2017JD026467 showed that in an equilibrium simulation the size distribution actually changed significantly even given equilibrium mean field statistics and smaller clouds coalesced and then split. How stable is Reff in your simulations over the time periods that the entrainment rate is diagnosed?

3) TKE and entrainment time: The conclusion section's take on cloud size vs. dilution is clear, and the results in the paper give a good indication about why correlations between cloud size and entraiment break down. I think a second paragraph, discussing in a similar way the impact of these results on assumptions underlying mixing-time parameterization schemes based on Neggers et al. 2002, like Tan et al. 2018 (doi: 10.1002/2017MS001162) would strengthen the conclusions.

---

## Author Comment (AC1) · 20 Sep 2020

We are very grateful to the two reviewers for their insightful and constructive comments, which have helped to substantially improve our analysis and physical interpretation. In the following, reviewer comments are written in black, author responses are written in blue, and passages of modified text in the manuscript are written in red.

**Referee 1**

General comments: This paper discusses the sensitivity of dilution of shallow convec-

tion as a function of large scale state. While the paper is not unique, it does yield another piece in the puzzle of figuring out how to parameterize entrainment. The paper is generally well written, but sometimes also a little descriptive, with the theoretical interpretation mainly hypothesized.

To enhance the paper, I have the following suggestions. I realize that not all of them may be feasible within the scope of this paper.

We thank the reviewer for their insightful and valuable comments.

1. The main finding to me is the dependence on 'continentality'. This is of course in reality more a dependence on the surface fluxes, and in itself it is no surprise that maritime clouds are different from continental once. So I would enjoy seeing this explored a bit further, for instance by looking into whether this is more an effect of the total buoyancy flux, or of the evaporative fraction/Bowen ratio. IN other words: Is this about the latent heat or the sensible heat?

   Indeed, it is the variation of the surface heat fluxes that explains the differing cloud-base mass fluxes ($m_b$) and, consequently, $\varepsilon$, between maritime and continental cloud fields (as mentioned in lines 364-367). To explore this relation more directly, we have conducted two sets of additional simulations, based around the ARM-SGP control case: the first varying the Bowen ratio ($\beta$) under fixed total surface heat flux (sensible plus latent), and the second varying sensible heat fluxes only, with latent-heat fluxes fixed to their control values.

   The results of both sets of experiments present a consistent picture of larger $m_b$ and consequentially weaker $\varepsilon$ for increased surface heating (Fig. 1). For the experiments with varying $\beta$, the sensitivity of both $m_b$ and $\varepsilon$ is weaker than for the HFX experiments, likely because the corresponding variations in subcloud turbulence intensity were partly compensated by variations in the subcloud humidity. For example, when $\beta$ was decreased, weaker subcloud turbulence, and hence reduced vertical displacement, was, in part, compensated by increased subcloud

humidity, which reduced the amount of vertical displacement required to reach the LFC. Thus, for the $\beta$ experiments, a stronger compensation mutes the sensitivity of $m_b$ and $\varepsilon$. These findings suggest that it is primarily $H$ that explains the sensitivity to continentality.

We have included two new figures focusing on the HFX experiments in the manuscript, the first showing time series (Fig. 2; Fig. 12 of the revised manuscript) and the second showing the relationship of sensible heat flux, cloud-base mass flux ($m_b$), and dilution rate (Fig. 3; Fig. 13 of the revised manuscript). We are grateful to the reviewer for this comment, which helped us to clarify the key aspect of continental environments that most directly influences $m_b$ and, hence, $\varepsilon$. The text accompanying these figures is provided below and on ll. 376-393 of the revised manuscript:

To further explore the sensitivity to surface heating, we conduct additional sensitivity experiments with modified sensible heat fluxes, based around the CTRL ARM-SGP case. These additional experiments are conducted without background wind to isolate the impact of buoyancy-driven, rather than shear-driven, turbulence on the dilution rate. In the first set of experiments, we vary the Bowen ratio $\beta$ (the ratio of sensible to latent heat fluxes) by 25% and 50% above and below its control values, while keeping $H + LE$ fixed to the CTRL value. For the second set of experiments, we hold the $LE$ fixed to its CTRL value and only change $H$ by 25% and 50% above and below the control values. The results of both sets of experiments present a consistent picture of larger $m_b$ and, consequentially, weaker $\varepsilon$ for increased surface heating. However, the sensitivity of both $m_b$ and $\varepsilon$ to surface hear flux changes in the $\beta$ experiments were found to be weaker than in the HFX experiments (not shown). This can be explained by a stronger compensation of the corresponding variations in subcloud turbulence intensity by variations in the subcloud humidity. For example, when $\beta$ is decreased, weaker subcloud turbulence, and hence reduced vertical displacement, is, in part, compensated by increased subcloud humidity, which reduces the amount of vertical displacement required to reach the LFC.

For the HFX experiments, decreasing $H$ tends to reduce the subcloud turbulence while increasing the subcloud specific humidity, with an attendant lowering of the cloud-base. These effects, which are shown for the case with a 50% reduction in $H$ (RHFX50) in Figs. 2j and h, are not unlike those arising from decreased $\beta$. However, their cancellation is weaker—the changes in turbulence intensity dominate. Weaker subcloud updrafts are less able to breach the LFC, leading to decreased $m_b$ and, in turn, increased $\varepsilon$ (Fig. 3). Hence, these findings suggest that it is primarily $H$ that explains the sensitivity to continentality, which is captured by the $m_b$ sensitivity in the TKE scaling.

2. The finding that the cloud base mass flux can explain the difference between the two regimes agrees with Dawe and Austin, and conflicts with Romps's Nature vs Nurture concept. Some discussion of that would help here.

Thank you for this suggestion. We agree that the placement of our findings in the context of the nature vs nurture debate of Romps and Kuang (2010) and Dawe and Austin (2012) had not been addressed properly in the original manuscript. In response, we have added the following paragraph discussing these concepts to the Discussion section (ll. 438-449):

The importance of $m_b$ on dilution reflects the broader importance of subcloud dynamics on cloud-layer convection, a crucial link that is being increasingly recognized (e.g., Tang and Kirshbaum, 2020). Both the initial cloud properties near cloud base (nature) and the environmental conditions experienced by the cloud as it ascends (nurture) have been examined for their roles in cloud evolution (e.g., Dawe and Austin, 2012; Romps and Kuang, 2010; Rousseau-Rizzi et al., 2017). Defining nature as the thermodynamic and kinematic state of a cloudy parcel at cloud base, Romps and Kuang (2010) analysed the relative importance of nature vs nurture from a parcel perspective. In LESs of shallow cumuli, they

found only a very weak correlation between the parcel's cloud-layer properties and its initial conditions at cloud base. They thus concluded that nature is of secondary importance for cloud evolution. In contrast, Dawe and Austin (2012) considered thermodynamic conditions as well as morphological characteristics of whole cloud entities as nature. While nurture primarily regulated the cloud thermodynamic properties, nature played an important role in controlling the cloud width and height in the upper cloud layer. Our results are consistent with Dawe and Austin (2012) in that cloud-base conditions may leave an imprint on the cloud properties above.

3. Detrainment is at least as important for cloud evolution as entrainment is. Is there a reason to barely include detrainment in this paper?

We agree that detrainment is an integral part of the cloud-environment mixing process, but investigating the sensitivity of both detrainment and entrainment/dilution to environmental conditions was not feasible within the scope of this study. To keep the manuscript to a reasonable length, we focused on the dilution problem. Note that detrainment was not completely neglected: analysis of detrainment was required to interpret the RH sensitivity. Although improving the understanding of detrainment is critical, it must be deferred to a subsequent study. In response to this comment, we have added some discussion on the importance of detrainment in the manuscript's conclusion (ll. 538-544):

For brevity, the focus of this study was placed on the sensitivity of cloud dilution to environmental conditions. However, since entrainment and dilution relate primarily to the inflow of surrounding air into the cloud, their counterpart—cloud outflow and detrainment—demand further analysis. de Rooy and Siebesma (2008) found detrainment to be sensitive to two environmental factors: cloud-layer depth and relative humidity. In more humid environments, entrainment of environmental air leads to less evaporative cooling, less buoyancy reversal, and hence less detrainment. Our cloud-layer RH results agree well with this finding. Nevertheless,

a more complete study of the sensitivity of detrainment to environmental conditions remains outstanding and is deferred to future work.

4. The cloud depth discussion is a bit too far simplified, as most clouds in a shallow Cu distribution would not come close to the cloud layer top, and therefore would not "feel" the extended depth of the layer. So what happens if you only sample clouds that actually did make it to the cloud layer top? Is there also some response in other variables here? Think of cloud (core) fraction, fluxes, etc.

Indeed, the majority of clouds do not reach the cloud-layer top at any given time, but this does not necessarily imply that the shallower clouds do not "feel" the layer depth. Clouds are influenced by the fluid both below and above them (and to the sides), and shallower ones may ultimately reach the cloud top at a later time.

The reviewer's comment poses a more general question: under what conditions can two sensitivity tests with differing cloud-layer depths be directly compared on equal footing? We argue that the key determinant is whether the *distribution* of normalized cloud depths (cloud depth normalized by cloud-layer depth) is similar between them. Similar distributions reflect dynamic similarity between the experiments, in that the cloud populations have similar success in reaching any given (normalized) height. By contrast, changes in the normalized cloud-depth distribution imply dynamic dissimilarity, which must be accounted for (possibly through the type of filtering exercise the reviewer suggests) to avoid a misleading interpretation.

To address this question, we have calculated the probability density function (PDF) of normalized cloud depths in each cloud-layer-depth sensitivity test (Fig. 4b). These distributions are similar in the different cases, indicating a large degree of dynamic similarity between them. We thus conclude that our evaluation method (studying all clouds, with no filtering) is not biased by differences in the ability of clouds to ascend through the layer. In other words, the clouds in the three cases all "feel" their layer depths to similar degrees.

To convey this finding in the manuscript, we have added Fig. 4b as Fig. 6b of the revised manuscript and expanded the text of the cloud-layer depth experiments as follows (ll. 292-300):

In any shallow cloud ensemble, the clouds may exhibit a wide range of depths at any instant, with most cloud tops falling well below the cloud-layer top. The distribution of individual cloud depths, normalized by the cloud-layer depth, is a morphological property that can be compared between different cases to assess their level of dynamic similarity. Systematic differences in these distributions would indicate that individual cloud depths do not simply scale with the cloud-layer depth (in a statistical sense), which could complicate a direct comparison of cloud dilution between them. To evaluate the similarity of the cloud ensembles for the current sensitivity tests, we compare their normalized-cloud-depth probability density functions (PDF) in Fig. 4b. At any given time, the vast majority of clouds have depths ($z_\mathrm{d}$) of less than half the cloud-layer depth ($\overline{z}_\mathrm{d}$), and very few reach the cloud-layer top. The distribution of normalized cloud depths is similar in all four cases, suggesting that these cloud ensembles can be directly compared on equal footing.

Concerning the reviewer's suggestion to filter cloud fields based on cloud depth, the similar normalized cloud-depth distributions among the cases does not suggest that such an exercise would be worthwhile. Moreover, given the small fraction of clouds that reach the cloud-layer top, restricting our analysis to just those clouds would reduce the sample size dramatically and thus undermine the statistical robustness. Recalculating $\varepsilon$ based on those clouds alone would generate dilution profiles that are even noisier than the ones shown in Fig. 4c, which would likely obscure any modest dilution sensitivities in these experiments.

Nevertheless, to respond to this comment as diligently as possible, and as a further validation of our standard evaluation method (considering all clouds), we have chosen to filter the clouds using a high-pass filter ($z_\mathrm{d} > 0.5\ \overline{z}_\mathrm{d}$) and have

recalculated the dilution rate for the remaining (deeper) cumuli (Fig. 5). This filtering leads a $\sim$6 % decrease in dilution rate for a doubling of $(z_{cld})_0$, compared to the $\sim$3 % reduction in dilution when all clouds are considered. Although the sensitivity doubles in a relative sense, the absolute sensitivity remains very weak. We are thus confident that our standard dilution calculation is not strongly biased by dynamic dissimilarities in the cloud fields, or the consideration of all clouds in the calculation. Because the analysis in Fig. 5 does not change our conclusions, we have left it out of the manuscript for the sake of brevity.

We have also calculated the distribution of the cloud depths relative to the layer boundaries for the subcloud-layer depth experiments (Fig. 6b; included as Fig. 8b in the revised manuscript). Clouds in all cases show a similar ability to ascend through the cloud layer, and the varying cloud-base heights do not result in different normalized cloud-depth distributions. The text accompanying the figure is included in our response to the reviewer's fifth comment and on lines 326-345.

5. Similarly, the subcloud layer alteration is a serious disturbance to the flow, and it seems like we are merely looking at the transient here. I am not sure I am learning a lot from that, so I recommend either removing the section, or clarifying its value.

We agree that some additional analysis is needed to strengthen this section and clarify its value. First recall that, in these simulations, the large-scale forcing profile was stretched or squashed from the control case (CTRL) to minimize the degree of flow disequilibrium. Nevertheless, some of the cases are steadier than others. As shown by time series of cloud-base height ($z_b$) in Fig. 7, the subcloud layer in SCL-SHAL1 deepens noticeably (80 m) over the course of the analysis period. In contrast, CTRL and SCL-DEEP1 exhibit relatively constant $z_b$ throughout the analysis period.

Given the relatively transient nature of the subcloud layer in SCL-SHAL1, we feel it important to acknowledge this transience and the potential confusion it might

cause. To that end, we have modified the description in lines 326-345 as follows:

Despite the adjustments to the large-scale forcing profiles to minimize the degree of disequilibrium in these cases, one of the two sensitivity tests (SCL-SHAL1) exhibits noticeable transience during the analysis period. Its cloud-base height increases from its initial, prescribed value of 250 m to an average of 350 m over the analysis period (not shown). By contrast, the cloud-base heights for CTRL and SCL-DEEP1 remain nearly fixed at their respective initial values of 500 m and 750 m. Although, for the sake of completeness, we show the results of SCL-SHAL1 in our subsequent analysis, its more transient nature may lead to a lack of robustness.

As before for the cloud-layer-depth experiments, we compare PDFs of normalized cloud depth for these three cases (Fig. 6b). The similar distributions thus produced suggests that the cloud ensembles are dynamically similar and can be straightforwardly compared. Near cloud base, the diagnosed $\varepsilon_{SC95}$ modestly but systematically decreases as the subcloud-layer depth is increased, while the value near cloud top remains similar (Fig. 6c). The layer-averaged $\varepsilon_{SC95}$ decreases by a total of about 15% for the near-tripling of the subcloud-layer depth between SCL-SHAL1 and SCL-DEEP1 (Fig. 9a). Although the transient SCL-SHAL1 case must be interpreted with caution, comparison between it and the CTRL case produces a similar trend as that found between CTRL and SCL-DEEP1.

As before, we use the TKE theory, as embodied in Eq. (8), to physically interpret the results. This theory reasonably captures the modest sensitivity of $\varepsilon_{SC95}$ to subcloud-layer depth (Fig. 9a), even for the transient SCL-SHAL1 case. Similar to the offsetting tendencies in Sect. 3.2.1, a ~5% increase of $z_{cld}$ is compensated by a 5% increase of CAPE$^{1/3}$ for the CTRL and SCL-DEEP1 experiments (Figs. 9c, d and e). For its part, $m_b$ tends to increase with subcloud-layer depth (Fig. 9b),

possibly owing to stronger, less hydrostatic turbulence in deeper subcloud layers (e.g., Tang and Kirshbaum, 2020). With offsetting effects on $z_{cld}$ and CAPE, the modest increase of $m_b$ in deeper subcloud layers explains a modest reduction in $\varepsilon_{TKE}$. An elaboration on the physical link between $\varepsilon_{TKE}$ and $m_b$ is provided below.

6. For the change in cloud layer humidity, the argument is that changes are based on the relative difference between environment and cloud, and by extension between environment and sub-cloud. If it is the relative difference, does that mean that it is only the gradient that matters? So if one would shift the entire profile, no response would be visible?

The reviewer's question is insightful but we cannot conclusively answer it with the available data. Intuitively, we would say yes, the difference between subcloud and cloud-layer specific humidity is much more important than the specific humidity itself. However, the strong sensitivity of saturation vapour pressure to temperature may induce a nonnegligible sensitivity to the absolute specific humidity too. This is because critical mixing fraction for buoyancy reversal may depend on temperature. At higher temperatures (and thus higher specific humidities), a given mixing fraction may be more prone to buoyancy reversal, due to increased evaporative cooling. From a buoyancy-sorting perspective, this would lead to a preference for detrainment rather than entrainment.

The above, however, is pure speculation, and providing a convincing answer would require a suite of new simulations that would substantially lengthen the manuscript. Because the manuscript is already quite long, and this issue does not threaten any of our key conclusions, we have opted not to address it herein.

**Referee 2**

This is interesting, carefully done investigation into the impact of environmental relative humidity, cloud and subcloud layer depth, and surface sensible and latent heat fluxes

on mixing and dilution in shallow clouds. The use of TKE scaling arguments allow the authors to bring some clarity into the broad range of effects that the environment has on cumulus mixing.

We thank the reviewer for their time and the useful comments.

Minor comments:

1. Resolution: I wasn't clear what the meaning of the parenthetical 100(50) notation was regarding $\Delta x$ and $\Delta y$ in Table 1. Does the 50 indicate that the same run was done at that finer resolution, and the results didn't change? It would be helpful to clarify this, and if possible to confirm that parameters like the cloud base mass flux didn't change with changing resolution.

   Thank you for pointing out the confusing formulation regarding the grid spacing of our simulations. We have clarified the description of the resolution in the manuscript and add the following explanation (ll. 139-141):

   Most simulations are conducted with the grid spacings mentioned in the reference literature (ranging from 64 m to 150 m). Additional high-resolution simulations with double the resolution have been conducted (indicated by the parenthetic grid spacings in Table 1).

2. Cloud size distribution: Satellite observations (e.g., Zhao and Di Girolamo, 2007, doi: 10.1029/2006 JD007371) indicate that cloud sizes follow a power-law distribution, so that the simple arithmetic mean isn't particularly representative of the actual size pdf. Feingold et al. (2017), doi: 10.1002/2017JD026467 showed that in an equilibrium simulation the size distribution actually changed significantly even given equilibrium mean field statistics and smaller clouds coalesced and then split. How stable is Reff in your simulations over the time periods that the entrainment rate is diagnosed?

The referee raises two valid points: (i) the validity of the choice of the arithmetic mean to characterize the observed cloud-size distributions and (ii) the temporal variability of the effective cloud radius ($R_{\mathrm{eff}}$) over the analysis period. Concerning the first issue, we first note that the cloud sizes are only analyzed qualitatively, so we only seek a metric that broadly captures the cloud-size distribution. We have calculated the cloud-size distribution for the cloud-layer and subcloud-layer depth experiments (Fig. 8; Fig. 14 in the revised manuscript). The different experiments exhibit similarly shaped distributions, and thus shifts in these distributions should be reasonably captured by their arithmetic means. As an additional evaluation, we have also compared profiles of the median cloud radius, which behaves very similarly to the mean (c.f. Figs. 8c-d and 9c-d). A discussion of the cloud-size distribution and arithmetic mean has been added to the manuscript (ll. 400-408):

Satellite observations (e.g., Zhao and Di Girolamo, 2007) and LES studies (e.g., Neggers et al., 2003) have shown that in shallow-cumulus cloud fields, the vast majority of clouds are small, and larger clouds are few and far-between. The cloud-size distribution has been variously characterized by lognormal, exponential, or power-law functions (e.g., Neggers et al., 2019). The $R_{\mathrm{eff}}$ distributions at the cloud-layer midpoints for the layer-depth sensitivity experiments of Sect. 3.2.1 and Sect. 3.2.2 reveal a similar pattern, with many small and few larger cumuli, broadly resembling lognormal functions (Fig. 8a and b). Because these distributions are similarly shaped, their arithmetic means should provide an adequate reflection of their statistical differences. For the cloud-layer depth sensitivity experiments, the distributions are nearly identical, and so are their arithmetic means (Fig. 8c). In contrast, the subcloud-layer depth sensitivity experiments exhibit a slight shift in the $R_{\mathrm{eff}}$ distribution toward larger values, which is again reflected in the mean profiles (Fig. 8d).

As for the time-variability of $R_{\mathrm{eff}}$, we have calculated time series of $R_{\mathrm{eff}}$ over the analysis period of the CTRL BOMEX and ARM-SGP experiments (Fig. 10),

with each point representing a 30-minute average. Although $R_{\text{eff}}$ fluctuates significantly (suggesting a pulsing behavior), particularly for BOMEX, there is no systematic trend in $R_{\text{eff}}$ over the analysis period. While this time-variability in $R_{\text{eff}}$ is certainly interesting and worthy of scientific investigation, its presence does not undermine our key conclusions, none of which rely on cloud-width arguments. Thus, for brevity, we have chosen not to include this analysis in the revised manuscript.

3. TKE and entrainment time: The conclusion section's take on cloud size vs. dilution is clear, and the results in the paper give a good indication about why correlations between cloud size and entrainment break down. I think a second paragraph, discussing in a similar way the impact of these results on assumptions underlying mixing-time parameterization schemes based on Neggers et al. (2002), like Tan et al. (2018), doi: 10.1002/2017MS001162 would strengthen the conclusions.

Thank you for the useful suggestion to explore the relationship between dilution and updraft vertical velocity. Although a more thorough discussion of the topic is included in a companion paper (Environmental sensitivities of shallow-cumulus dilution. Part II: Vertical wind profile—to be submitted soon), we have compared the simulated dilution rates to corresponding predictions based on vertical velocity and buoyancy (Tan et al., 2018) in Fig. 11 (included as Fig. 17 in the revised manuscript). Their relation explains a substantial fraction of the variability seen in the different experiments, but still leaves much to be desired. A paragraph accompanying the figure and discussing the general relation between dilution and vertical velocity is included in the lines 475-486.

Neggers et al. (2002) developed a multiparcel entrainment model for shallow cumulus convection, in which dilution was prescribed to be inversely proportional to the vertical velocity ($w$). The reasoning behind this sensitivity is that, for a faster ascending air parcel, entrainment has less time to dilute the cloudy parcel than

for a slower rising one. Subsequent studies have supported these findings and formulated more complex relationships between core properties and $\varepsilon$. Tan et al. (2018), for example, parameterized $\varepsilon$ using a combination of cloud buoyancy ($b$) and $w$:

$$\varepsilon_{\mathsf{Tan}} = c_\varepsilon \frac{\max(0, b)}{w^2} \ . \tag{1}$$

We calculate $\varepsilon_{\mathsf{Tan}}$ using bulk core statistics and compare it to the calculated $\varepsilon_{\mathsf{SC95}}$. With the coefficient $c_\varepsilon$ set to 0.3 (instead of 0.12 as suggested by Tan et al. (2018)), $\varepsilon_{\mathsf{Tan}}$ captures the overall trend of larger $\varepsilon$ in maritime clouds and smaller $\varepsilon$ in continental clouds (Fig. 11). However, this relation cannot explain all of the sensitivities found in the experiments. For example, the slightly larger $\varepsilon$ in RICO, relative to BOMEX, is not captured, and the differences between ARM-SGP and RACORO are over-predicted. Thus, additional factors beyond $b$ and $w$ may be required to more accurately represent the sensitivity of cloud dilution to environmental conditions.

**References**

Dawe, J. T. and P. H. Austin, 2012: Statistical analysis of an LES shallow cumulus cloud ensemble using a cloud tracking algorithm. *Atmos. Chem. Phys.*, **12**, 1101–1119, doi: 10.5194/acp-12-1101-2012.

de Rooy, W. C. and A. P. Siebesma, 2008: A Simple Parameterization for Detrainment in Shallow Cumulus. *Mon. Wea. Rev.*, **136**, 560–576, doi:10.1175/2007MWR2201.1.

Feingold, G., J. Balsells, F. Glassmeier, T. Yamaguchi, J. Kazil, and A. McComiskey, 2017: Analysis of albedo versus cloud fraction relationships in liquid water clouds using heuristic models and large eddy simulation. *J. Geophys. Res. Atmos.*, **122**, 7086–7102, doi:10.1002/2017JD026467.

Neggers, R. A. J., P. J. Griewank, and T. Heus, 2019: Power-Law Scaling in the Internal Variability of Cumulus Cloud Size Distributions due to Subsampling and Spatial Organization . *J. Atmos. Sci.*, **76**, 1489–1503, doi:10.1175/JAS-D-18-0194.1.

Neggers, R. A. J., H. J. J. Jonker, and A. P. Siebesma, 2003: Size Statistics of Cumulus Cloud Populations in Large-Eddy Simulations. *J. Atmos. Sci.*, **60**, 1060–1074, doi: 10.1175/1520-0469(2003)60<1060:SSOCCP>2.0.CO;2.

Neggers, R. A. J., A. P. Siebesma, and H. J. J. Jonker, 2002: A Multiparcel Model for Shallow Cumulus Convection. *J. Atmos. Sci.*, **59**, 1655–1668, doi:10.1175/1520-0469(2002)059<1655: AMMFSC>2.0.CO;2.

Romps, D. M. and Z. Kuang, 2010: Nature versus Nurture in Shallow Convection. *J. Atmos. Sci.*, **67**, 1655–1666, doi:10.1175/2009JAS3307.1.

Rousseau-Rizzi, R., D. J. Kirshbaum, and M. K. Yau, 2017: Initiation of Deep Convection over an Idealized Mesoscale Convergence Line. *J. Atmos. Sci.*, **74**, 835–853, doi: 10.1175/JAS-D-16-0221.1.

Tan, Z., C. M. Kaul, K. G. Pressel, Y. Cohen, T. Schneider, and J. Teixeira, 2018: An extended eddy-diffusivity mass-flux scheme for unified representation of subgrid-scale turbulence and convection. *J. Adv. Mod. Earth Sys.*, **10**, 770–800, doi:10.1002/2017MS001162.

Tang, S. L. and D. J. Kirshbaum, 2020: On the sensitivity of deep-convection initiation to horizontal grid resolution. *Quart. J. Roy. Meteor. Soc.*, **146**, 1085–1105, doi:10.1002/qj.3726.

Zhao, G. and L. Di Girolamo, 2007: Statistics on the macrophysical properties of trade wind cumuli over the tropical western Atlantic. *J. Geophys. Res.*, **112**, D10 204, doi: 10.1029/2006JD007371.

[Figure]

**Fig. 1.** The relation of sensible heat flux/Bowen ratio, cloud-base mass flux, and dilution rate for the Bowen ratio and HFX experiments as well as BOMEX and ARM-SGP.

BOMEX

ARM-SGP

(a)

(b)

LE

H

(c)

(d)

(e)

(f)

(g)

(h)

(i)

(j)

Time

Local Time

**Fig. 2.** Time series for BOMEX (first column), ARM-SGP and ARM-SGP-RHFX50 (second column) experiments. The dashed line in the second column indicates the RHFX50 experiment.

[Figure]

[Figure]

**Fig. 3.** The relation of sensible heat flux, cloud-base mass flux, and dilution rate for BOMEX, ARM-SGP and ARM-SGP-HFX experiments.

[Figure]

**Fig. 4.** (a) Initial profiles for the experiments with different cloud-layer depth. (b) The PDF of individual cloud depth normalized by the cloud-layer depth and (c) the fractional dilution rate profiles.

**Fig. 5.** The dilution rate averaged over the central 50% of the cloud layer for all clouds compared to the dilution for clouds that have cloud depths normalized by the cloud-layer depth larger than 0.5.

[Figure]

**Fig. 6.** Same as Fig. 4 but for the different subcloud-layer-depth experiments.

[Figure]

**Fig. 7.** Cloud-base height for the subcloud-layer depth experiments.

[Figure]

**Fig. 8.** PDF of the cloud radius for (a) cloud-layer depth and (b) subcloud-layer depth experiments. The averaged cloud radius profiles for the same cases are shown in (c) and (d), respectively.

**Cloud-layer depth**

(a)

CL-DEEP2
CL-DEEP1
CTRL
CL-SHAL1

**Subloud-layer depth**

(b)

SCL-DEEP1
CTRL
SCL-SHAL1

(c)

(d)

**Fig. 9.** PDF of the cloud radius for (a) cloud-layer depth and (b) subcloud-layer depth experiments. The median cloud radius profiles for the same cases are shown in (c) and (d), respectively.

**Fig. 10.** Time series of the effective radius (averaged over 30-minute center around the indicated time into the analysis period) for CTRL BOMEX and ARM-SGP experiments.

**Fig. 11.** Relationship between dilution rate calculated using Eq. (1) and dilution rates approximated using Eq. (26) in Tan et al., 2018 for all 18 experiments.

---

## Author Response (AR2)

**Responses to referee comments, Atmos. Chem. Phys. Discuss. acp-2020-336**

We are very grateful to the two reviewers for their insightful and constructive comments, which have helped to substantially improve our analysis and physical interpretation. In the following, reviewer comments are written in black, author responses are written in blue, and passages of modified text are written in red.

**Referee 1**

General comments: This paper discusses the sensitivity of dilution of shallow convection as a function of large scale state. While the paper is not unique, it does yield another piece in the puzzle of figuring out how to parameterize entrainment. The paper is generally well written, but sometimes also a little descriptive, with the theoretical interpretation mainly hypothesized.

To enhance the paper, I have the following suggestions. I realize that not all of them may be feasible within the scope of this paper.

We thank the reviewer for their insightful and valuable comments.

1. The main finding to me is the dependence on 'continentality'. This is of course in reality more a dependence on the surface fluxes, and in itself it is no surprise that maritime clouds are different from continental once. So I would enjoy seeing this explored a bit further, for instance by looking into whether this is more an effect of the total buoyancy flux, or of the evaporative fraction/Bowen ratio. IN other words: Is this about the latent heat or the sensible heat?

   Indeed, it is the variation of the surface heat fluxes that explains the differing cloud-base mass fluxes ($m_{\mathrm{b}}$) and, consequently, $\varepsilon$, between maritime and continental cloud fields (as mentioned in lines 364-367). To explore this relation more directly, we have conducted two sets of additional simulations, based around the ARM-SGP control case: the first varying the Bowen ratio ($\beta$) under fixed total surface heat flux (sensible plus latent), and the second varying sensible heat fluxes only, with latent-heat fluxes fixed to their control values.

   The results of both sets of experiments present a consistent picture of larger $m_{\mathrm{b}}$ and consequentially weaker $\varepsilon$ for increased surface heating (Fig. 1). For the experiments with varying $\beta$, the sensitivity of both $m_{\mathrm{b}}$ and $\varepsilon$ is weaker than for the HFX experiments, likely because the corresponding variations in subcloud turbulence intensity were partly compensated by variations in the subcloud humidity. For example, when $\beta$ was decreased, weaker subcloud turbulence, and hence reduced vertical displacement, was, in part, compensated by increased subcloud humidity, which reduced the amount of vertical displacement required to reach the LFC. Thus, for the $\beta$ experiments, a stronger compensation mutes the sensitivity of $m_{\mathrm{b}}$ and $\varepsilon$. These findings suggest that it is primarily $H$ that explains the sensitivity to continentality.

   We have included two new figures focusing on the HFX experiments in the manuscript, the first showing time series (Fig. 2; Fig. 12 of the revised manuscript) and the second showing relationship of sensible heat flux, cloud-base mass flux ($m_{\mathrm{b}}$), and dilution rate (Fig. 3; Fig. 13 of the revised manuscript). We are grateful to the reviewer for this comment, which helped us to clarify the key aspect of continental environments that most directly influences $m_{\mathrm{b}}$ and, hence, $\varepsilon$. The text accompanying these figures is provided below and on ll 376-393 of the revised manuscript:

   To further explore the sensitivity to surface heating, we conduct additional sensitivity experiments with modified sensible heat fluxes, based around the CTRL ARM-SGP case. These additional experiments are conducted without background wind to isolate the impact of buoyancy-driven, rather than shear-driven, turbulence on the dilution rate. In the first set of experiments, we vary the Bowen ratio $\beta$ (the ratio of sensible to latent heat fluxes) by 25% and 50% above and below its control values, while keeping $H + LE$ fixed to the CTRL value. For the second set of experiments,

[Figure]

Figure 1: (a) and (b) The cloud-base mass flux ($m_b$) as a function of the sensible heat flux ($H$) and Bowen ratio ($\beta$) and (c) and (d) the dilution rate ($\varepsilon$) averaged over the central 50% of the cloud layer as a function of $m_b$, for BOMEX and ARM-SGP as well as the ARM-SGP-$\beta$ experiments (first column) and ARM-SGP-HFX experiments (second column).

we hold the $LE$ fixed to its CTRL value and only change $H$ by 25% and 50% above and below the control values. The results of both sets of experiments present a consistent picture of larger $m_b$ and, consequentially, weaker $\varepsilon$ for increased surface heating. However, the sensitivity of both $m_b$ and $\varepsilon$ to surface hear flux changes in the $\beta$ experiments were found to be weaker than in the HFX experiments (not shown). This can be explained by a stronger compensation of the corresponding variations in subcloud turbulence intensity by variations in the subcloud humidity. For example, when $\beta$ is decreased, weaker subcloud turbulence, and hence reduced vertical displacement, is, in part, compensated by increased subcloud humidity, which reduces the amount of vertical displacement required to reach the LFC.

For the HFX experiments, decreasing $H$ tends to reduce the subcloud turbulence while increasing the subcloud specific humidity, with an attendant lowering of the cloud-base. These effects, which are shown for the case with a 50% reduction in $H$ (RHFX50) in Figs. 2j and h, are not unlike those arising from decreased $\beta$. However, their cancellation is weaker—the changes in turbulence intensity dominate. Weaker subcloud updrafts are less able to breach the LFC, leading to decreased $m_b$ and, in turn, increased $\varepsilon$ (Fig. 3). Hence, these findings suggest that it is primarily $H$ that explains the sensitivity to continentality, which is captured by the $m_b$ sensitivity in the TKE scaling.

[Figure]

Figure 2: Time series of (a-b) latent ($LE$) and sensible ($H$) heat fluxes, (c-d) total cloud cover, (e-f) cloud-base and cloud-top height ($z_b$ and $z_t$), (g-h) cloud-core-base mass flux, and (i-j) vertically integrated TKE. The first column shows the time series for the BOMEX simulations, and the second column depicts the temporal evolution of the ARM-SGP as well as the diurnal cycle of the ARM-SGP experiments with a reduced $H$ by 50% (RHFX50) in dashed lines. The gray shading indicates the time window over which averages have been performed.

2. The finding that the cloud base mass flux can explain the difference between the two regimes agrees with Dawe and Austin, and conflicts with Romps's Nature vs Nurture concept. Some discussion of

[Figure]

Figure 3: (a) The cloud-base mass flux ($m_b$) as a function of sensible heat flux ($H$) and (b) the dilution rate ($\varepsilon$) averaged over the central 50% of the cloud layer as a function of $m_b$ for BOMEX and ARM-SGP as well as the ARM-SGP-HFX experiments.

that would help here.

Thank you for this suggestion. We agree that the placement of our findings in the context of the nature vs nurture debate of Romps and Kuang (2010) and Dawe and Austin (2012) had not been addressed properly in the original manuscript. In response, we have added the following paragraph discussing these concepts to the Discussion section (ll. 438-449):

The importance of $m_b$ on dilution reflects the broader importance of subcloud dynamics on cloud-layer convection, a crucial link that is being increasingly recognized (e.g., Tang and Kirshbaum, 2020). Both the initial cloud properties near cloud base (nature) and the environmental conditions experienced by the cloud as it ascends (nurture) have been examined for their roles in cloud evolution (e.g., Dawe and Austin, 2012; Romps and Kuang, 2010; Rousseau-Rizzi et al., 2017). Defining nature as the thermodynamic and kinematic state of a cloudy parcel at cloud base, Romps and Kuang (2010) analysed the relative importance of nature vs nurture from a parcel perspective. In LESs of shallow cumuli, they found only a very weak correlation between the parcel's cloud-layer properties and its initial conditions at cloud base. They thus concluded that nature is of secondary importance for cloud evolution. In contrast, Dawe and Austin (2012) considered thermodynamic conditions as well as morphological characteristics of whole cloud entities as nature. While nurture primarily regulated the cloud thermodynamic properties, nature played an important role in controlling the cloud width and height in the upper cloud layer. Our results are consistent with Dawe and Austin (2012) in that cloud-base conditions may leave an imprint on the cloud properties above.

3. Detrainment is at least as important for cloud evolution as entrainment is. Is there a reason to barely include detrainment in this paper?

We agree that detrainment is an integral part of the cloud-environment mixing process, but investigating the sensitivity of both detrainment and entrainment/dilution to environmental conditions was not feasible within the scope of this study. To keep the manuscript to a reasonable length, we focused on the dilution problem. Note that detrainment was not completely neglected: analysis of detrainment was required to interpret the RH sensitivity. Although improving the understanding of detrainment is critical, it must be deferred to a subsequent study. In response to this comment, we have added some discussion on the importance of detrainment in the manuscript's conclusion (ll. 538-544):

For brevity, the focus of this study was placed on the sensitivity of cloud dilution to environmental conditions. However, since entrainment and dilution relate primarily to the inflow of surrounding air into the cloud, their counterpart—cloud outflow and detrainment—demand further analysis. de Rooy and Siebesma (2008) found detrainment to be sensitive to two environmental factors: cloud-layer depth and relative humidity. In more humid environments, entrainment of environmental air leads to less evaporative cooling, less buoyancy reversal, and hence less detrainment. Our cloud-layer RH results agree well with this finding. Nevertheless, a more complete study of the sensitivity of detrainment to environmental conditions remains outstanding and is deferred to future work.

4. The cloud depth discussion is a bit too far simplified, as most clouds in a shallow Cu distribution would not come close to the cloud layer top, and therefore would not "feel" the extended depth of the layer. So what happens if you only sample clouds that actually did make it to the cloud layer top? Is there also some response in other variables here? Think of cloud (core) fraction, fluxes, etc.

Indeed, the majority of clouds do not reach the cloud-layer top at any given time, but this does not necessarily imply that the shallower clouds do not "feel" the layer depth. Clouds are influenced by the fluid both below and above them (and to the sides), and shallower ones may ultimately reach the cloud top at a later time.

The reviewer's comment poses a more general question: under what conditions can two sensitivity tests with differing cloud-layer depths be directly compared on equal footing? We argue that the key determinant is whether the *distribution* of normalized cloud depths (cloud depth normalized by cloud-layer depth) is similar between them. Similar distributions reflect dynamic similarity between the experiments, in that the cloud populations have similar success in reaching any given (normalized) height. By contrast, changes in the normalized cloud-depth distribution imply dynamic dissimilarity, which must be accounted for (possibly through the type of filtering exercise the reviewer suggests) to avoid a misleading interpretation.

To address this question, we have calculated the probability density function (PDF) of normalized cloud depths in each cloud-layer-depth sensitivity test (Fig. 4b). These distributions are similar in the different cases, indicating a large degree of dynamic similarity between them. We thus conclude that our evaluation method (studying all clouds, with no filtering) is not biased by differences in the ability of clouds to ascend through the layer. In other words, the clouds in the three cases all "feel" their layer depths to similar degrees.

To convey this finding in the manuscript, we have added Fig. 4b as Fig. 6b of the revised manuscript and expanded the text of the cloud-layer depth experiments as follows (ll. 292-300):

In any shallow cloud ensemble, the clouds may exhibit a wide range of depths at any instant, with most cloud tops falling well below the cloud-layer top. The distribution of individual cloud depths, normalized by the cloud-layer depth, is a morphological property that can be compared between different cases to assess their level of dynamic similarity. Systematic differences in these distributions would indicate that individual cloud depths do not simply scale with the cloud-layer depth (in a statistical sense), which could complicate a direct comparison of cloud dilution between them. To evaluate the similarity of the cloud ensembles for the current sensitivity tests, we compare their normalized-cloud-depth probability density functions (PDF) in Fig. 4b. At any given time, the vast majority of clouds have depths ($z_\mathrm{d}$) of less than half the cloud-layer depth ($\overline{z}_\mathrm{d}$), and very few reach the cloud-layer top. The distribution of normalized cloud depths is similar in all four cases, suggesting that these cloud ensembles can be directly compared on equal footing.

Concerning the reviewer's suggestion to filter cloud fields based on cloud depth, the similar normalized cloud-depth distributions among the cases does not suggest that such an exercise would be worthwhile. Moreover, given the small fraction of clouds that reach the cloud-layer top, restricting our analysis to just those clouds would reduce the sample size dramatically and thus undermine the statistical robustness. Recalculating $\varepsilon$ based on those clouds alone would generate dilution profiles

[Figure]

Figure 4: (a) Initial profiles of potential temperature ($\theta$) and water vapor mixing ratio ($q_v$) for the experiments with different cloud-layer depths. (b) The probability density function (PDF) of individual cloud depth normalized by the cloud-layer depth and (c) the fractional-dilution-rate profiles for the cloud-layer-depth experiments.

that are even noisier than the ones shown in Fig. 4c, which would likely obscure any modest dilution sensitivities in these experiments.

Nevertheless, to respond to this comment as diligently as possible, and as a further validation of our standard evaluation method (considering all clouds), we have chosen to filter the clouds using a high-pass filter ($z_d > 0.5\ \bar{z}_d$) and have recalculated the dilution rate for the remaining (deeper) cumuli (Fig. 5). This filtering leads a $\sim$6 % decrease in dilution rate for a doubling of $(z_{cld})_0$, compared to the $\sim$3 % reduction in dilution when all clouds are considered. Although the sensitivity doubles in a relative sense, the absolute sensitivity remains very weak. We are thus confident that our standard dilution calculation is not strongly biased by dynamic dissimilarities in the cloud fields, or the consideration of all clouds in the calculation. Because the analysis in Fig. 5 does not change our conclusions, we have left it out of the manuscript for the sake of brevity.

[Figure]

Figure 5: The fractional dilution rate ($\varepsilon$) averaged over the central 50 % of the cloud layer for all clouds compared to $\varepsilon$ for clouds that have cloud depths normalized by the cloud-layer depth larger than 0.5.

We have also calculated the distribution of the cloud depths relative to the layer boundaries for the subcloud-layer depth experiments (Fig. 6b; included as Fig. 8b in the revised manuscript). Clouds in all cases show a similar ability to ascend through the cloud layer, and the varying cloud-base heights do not result in different normalized cloud-depth distributions. The text accompanying the figure is included in our response to the reviewer's fifth comment and on lines 326-345.

[Figure]

Figure 6: Same as Fig. 4 but for the different subcloud-layer-depth experiments.

5. Similarly, the subcloud layer alteration is a serious disturbance to the flow, and it seems like we are merely looking at the transient here. I am not sure I am learning a lot from that, so I recommend either removing the section, or clarifying its value.

We agree that some additional analysis is needed to strengthen this section and clarify its value. First recall that, in these simulations, the large-scale forcing profile was stretched or squashed from the control case (CTRL) to minimize the degree of flow disequilibrium. Nevertheless, some of the cases are steadier than others. As shown by time series of cloud-base height ($z_b$) in Fig. 7, the subcloud layer in SCL-SHAL1 deepens noticeably (80 m) over the course of the analysis period. In contrast, CTRL and SCL-DEEP1 exhibit relatively constant $z_b$ throughout the analysis period.

[Figure]

Figure 7: Cloud-base height ($z_b$) for the subcloud-layer depth experiments.

Given the relatively transient nature of the subcloud layer in SCL-SHAL1, we feel it important to acknowledge this transience and the potential confusion it might cause. To that end, we have modified the description in lines 326-345 as follows:

Despite the adjustments to the large-scale forcing profiles to minimize the degree of disequilibrium

in these cases, one of the two sensitivity tests (SCL-SHAL1) exhibits noticeable transience during the analysis period. Its cloud-base height increases from its initial, prescribed value of 250 m to an average of 350 m over the analysis period (not shown). By contrast, the cloud-base heights for CTRL and SCL-DEEP1 remain nearly fixed at their respective initial values of 500 m and 750 m. Although, for the sake of completeness, we show the results of SCL-SHAL1 in our subsequent analysis, its more transient nature may lead to a lack of robustness.

As before for the cloud-layer-depth experiments, we compare PDFs of normalized cloud depth for these three cases (Fig. 6b). The similar distributions thus produced suggests that the cloud ensembles are dynamically similar and can be straightforwardly compared. Near cloud base, the diagnosed $\varepsilon_{\mathrm{SC95}}$ modestly but systematically decreases as the subcloud-layer depth is increased, while the value near cloud top remains similar (Fig. 6c). The layer-averaged $\varepsilon_{\mathrm{SC95}}$ decreases by a total of about 15% for the near-tripling of the subcloud-layer depth between SCL-SHAL1 and SCL-DEEP1 (Fig. 9a). Although the transient SCL-SHAL1 case must be interpreted with caution, comparison between it and the CTRL case produces a similar trend as that found between CTRL and SCL-DEEP1.

As before, we use the TKE theory, as embodied in Eq. (8), to physically interpret the results. This theory reasonably captures the modest sensitivity of $\varepsilon_{\mathrm{SC95}}$ to subcloud-layer depth (Fig. 9a), even for the transient SCL-SHAL1 case. Similar to the offsetting tendencies in Sect. 3.2.1, a ~5% increase of $z_{\mathrm{cld}}$ is compensated by a 5% increase of $\mathrm{CAPE}^{1/3}$ for the CTRL and SCL-DEEP1 experiments (Figs. 9c, d and e). For its part, $m_{\mathrm{b}}$ tends to increase with subcloud-layer depth (Fig. 9b), possibly owing to stronger, less hydrostatic turbulence in deeper subcloud layers (e.g., Tang and Kirshbaum, 2020). With offsetting effects on $z_{\mathrm{cld}}$ and CAPE, the modest increase of $m_{\mathrm{b}}$ in deeper subcloud layers explains a modest reduction in $\varepsilon_{\mathrm{TKE}}$. An elaboration on the physical link between $\varepsilon_{\mathrm{TKE}}$ and $m_{\mathrm{b}}$ is provided below.

6. For the change in cloud layer humidity, the argument is that changes are based on the relative difference between environment and cloud, and by extension between environment and sub-cloud. If it is the relative difference, does that mean that it is only the gradient that matters? So if one would shift the entire profile, no response would be visible?

The reviewer's question is insightful but we cannot conclusively answer it with the available data. Intuitively, we would say yes, the difference between subcloud and cloud-layer specific humidity is much more important than the specific humidity itself. However, the strong sensitivity of saturation vapour pressure to temperature may induce a nonnegligible sensitivity to the absolute specific humidity too. This is because critical mixing fraction for buoyancy reversal may depend on temperature. At higher temperatures (and thus higher specific humidities), a given mixing fraction may be more prone to buoyancy reversal, due to increased evaporative cooling. From a buoyancy-sorting perspective, this would lead to a preference for detrainment rather than entrainment.

The above, however, is pure speculation, and providing a convincing answer would require a suite of new simulations that would substantially lengthen the manuscript. Because the manuscript is already quite long, and this issue does not threaten any of our key conclusions, we have opted not to address it herein.

**Referee 2**

This is interesting, carefully done investigation into the impact of environmental relative humidity, cloud and subcloud layer depth, and surface sensible and latent heat fluxes on mixing and dilution in shallow clouds. The use of TKE scaling arguments allow the authors to bring some clarity into the broad range of effects that the environment has on cumulus mixing.

We thank the reviewer for their time and the useful comments.

Minor comments:

1. Resolution: I wasn't clear what the meaning of the parenthetical 100(50) notation was regarding $\Delta x$ and $\Delta y$ in Table 1. Does the 50 indicate that the same run was done at that finer resolution, and the results didn't change? It would be helpful to clarify this, and if possible to confirm that parameters like the cloud base mass flux didn't change with changing resolution.

Thank you for pointing out the confusing formulation regarding the grid spacing of our simulations. We have clarified the description of the resolution in the manuscript and add the following explanation (ll. 139-141):

Most simulations are conducted with the grid spacings mentioned in the reference literature (ranging from 64 m to 150 m). Additional high-resolution simulations with double the resolution have been conducted (indicated by the parenthetic grid spacings in Table 1).

2. Cloud size distribution: Satellite observations (e.g., Zhao and Di Girolamo, 2007, doi: 10.1029/2006 JD007371) indicate that cloud sizes follow a power-law distribution, so that the simple arithmetic mean isn't particularly representative of the actual size pdf. Feingold et al. (2017), doi: 10.1002/2017JD02 6467 showed that in an equilibrium simulation the size distribution actually changed significantly even given equilibrium mean field statistics and smaller clouds coalesced and then split. How stable is Reff in your simulations over the time periods that the entrainment rate is diagnosed?

The referee raises two valid points: (i) the validity of the choice of the arithmetic mean to characterize the observed cloud-size distributions and (ii) the temporal variability of the effective cloud radius ($R_{\mathrm{eff}}$) over the analysis period. Concerning the first issue, we first note that the cloud sizes are only analyzed qualitatively, so we only seek a metric that broadly captures the cloud-size distribution. We have calculated the cloud-size distribution for the cloud-layer and subcloud-layer depth experiments (Fig. 8; Fig. 14 in the revised manuscript). The different experiments exhibit similarly shaped distributions, and thus shifts in these distributions should be reasonably captured by their arithmetic means. As an additional evaluation, we have also compared profiles of the median cloud radius, which behaves very similarly to the mean (c.f. Figs. 8c-d and 9c-d). A discussion of the cloud-size distribution and arithmetic mean has been added to the manuscript (ll. 400-408):

Satellite observations (e.g., Zhao and Di Girolamo, 2007) and LES studies (e.g., Neggers et al., 2003) have shown that in shallow-cumulus cloud fields, the vast majority of clouds are small, and larger clouds are few and far-between. The cloud-size distribution has been variously characterized by lognormal, exponential, or power-law functions (e.g., Neggers et al., 2019). The $R_{\mathrm{eff}}$ distributions at the cloud-layer midpoints for the layer-depth sensitivity experiments of Sect. 3.2.1 and Sect. 3.2.2 reveal a similar pattern, with many small and few larger cumuli, broadly resembling lognormal functions (Fig. 8a and b). Because these distributions are similarly shaped, their arithmetic means should provide an adequate reflection of their statistical differences. For the cloud-layer depth sensitivity experiments, the distributions are nearly identical, and so are their arithmetic means (Fig. 8c). In contrast, the subcloud-layer depth sensitivity experiments exhibit a slight shift in the $R_{\mathrm{eff}}$ distribution toward larger values, which is again reflected in the mean profiles (Fig. 8d).

As for the time-variability of $R_{\mathrm{eff}}$, we have calculated time series of $R_{\mathrm{eff}}$ over the analysis period of the CTRL BOMEX and ARM-SGP experiments (Fig. 10), with each point representing a 30-minute average. Although $R_{\mathrm{eff}}$ fluctuates significantly (suggesting a pulsing behavior), particularly for BOMEX, there is no systematic trend in $R_{\mathrm{eff}}$ over the analysis period. While this time-variability in $R_{\mathrm{eff}}$ is certainly interesting and worthy of scientific investigation, its presence does not undermine our key conclusions, none of which rely on cloud-width arguments. Thus, for brevity, we have chosen not to include this analysis in the revised manuscript.

[Figure]

Figure 8: PDF of the effective cloud radius ($R_{eff}$) halfway into the the respective cloud layers for the experiments with (a) different cloud-layer depths and (b) varying subcloud-layer depths. The averaged $R_{eff}$ profile for the same experiments are shown in (c) and (d), respectively.

3. TKE and entrainment time: The conclusion section's take on cloud size vs. dilution is clear, and the results in the paper give a good indication about why correlations between cloud size and entrainment break down. I think a second paragraph, discussing in a similar way the impact of these results on assumptions underlying mixing-time parameterization schemes based on Neggers et al. (2002), like Tan et al. (2018), doi: 10.1002/2017MS001162 would strengthen the conclusions.

Thank you for the useful suggestion to explore the relationship between dilution and updraft vertical velocity. Although a more thorough discussion of the topic is included in a companion paper (Environmental sensitivities of shallow-cumulus dilution. Part II: Vertical wind profile—to be submitted soon), we have compared the simulated dilution rates to corresponding predictions based on vertical velocity and buoyancy (Tan et al., 2018) in Fig. 11 (included as Fig. 17 in the revised

[Figure]

Figure 9: PDF of the effective cloud radius ($R_{\text{eff}}$) halfway into the the respective cloud layers for the experiments with (a) different cloud-layer depths and (b) varying subcloud-layer depths. The median $R_{\text{eff}}$ profile for the same experiments are shown in (c) and (d), respectively.

manuscript). Their relation explains a substantial fraction of the variability seen in the different experiments, but still leaves much to be desired. A paragraph accompanying the figure and discussing the general relation between dilution and vertical velocity is included in the lines 475-486.

Neggers et al. (2002) developed a multiparcel entrainment model for shallow cumulus convection, in which dilution was prescribed to be inversely proportional to the vertical velocity ($w$). The reasoning behind this sensitivity is that, for a faster ascending air parcel, entrainment has less time to dilute the cloudy parcel than for a slower rising one. Subsequent studies have supported these findings and formulated more complex relationships between core properties and $\varepsilon$. Tan et al.

[Figure]

Figure 10: Time series of the effective radius (averaged over 30-minute center around the indicated time into the analysis period) for CTRL BOMEX and ARM-SGP experiments

(2018), for example, parameterized $\varepsilon$ using a combination of cloud buoyancy ($b$) and $w$:

$$\varepsilon_{\text{Tan}} = c_\varepsilon \frac{\max(0, b)}{w^2} \ . \tag{1}$$

We calculate $\varepsilon_{\text{Tan}}$ using bulk core statistics and compare it to the calculated $\varepsilon_{\text{SC95}}$. With the coefficient $c_\varepsilon$ set to 0.3 (instead of 0.12 as suggested by Tan et al. (2018)), $\varepsilon_{\text{Tan}}$ captures the overall trend of larger $\varepsilon$ in maritime clouds and smaller $\varepsilon$ in continental clouds (Fig. 11). However, this relation cannot explain all of the sensitivities found in the experiments. For example, the slightly larger $\varepsilon$ in RICO, relative to BOMEX, is not captured, and the differences between ARM-SGP and RACORO are over-predicted.. Thus, additional factors beyond $b$ and $w$ may be required to more accurately represent the sensitivity of cloud dilution to environmental conditions.

[Figure]

Figure 11: Relationship between dilution rate calculated using Eq. (1) ($\varepsilon_{SC95}$) and dilution rates approximated using Eq. (26) in Tan et al. (2018) for all 18 experiments. The non-dimensional coefficients $c_\varepsilon$ has been set to 0.3.

Neggers, R. A. J., A. P. Siebesma, and H. J. J. Jonker, 2002: A Multiparcel Model for Shallow Cumulus Convection. *J. Atmos. Sci.*, **59**, 1655–1668, doi:10.1175/1520-0469(2002)059⟨1655:AMMFSC⟩2.0.CO;2.

Romps, D. M. and Z. Kuang, 2010: Nature versus Nurture in Shallow Convection. *J. Atmos. Sci.*, **67**, 1655–1666, doi:10.1175/2009JAS3307.1.

Rousseau-Rizzi, R., D. J. Kirshbaum, and M. K. Yau, 2017: Initiation of Deep Convection over an Idealized Mesoscale Convergence Line. *J. Atmos. Sci.*, **74**, 835–853, doi:10.1175/JAS-D-16-0221.1.

Tan, Z., C. M. Kaul, K. G. Pressel, Y. Cohen, T. Schneider, and J. Teixeira, 2018: 
[revised manuscript text omitted]